# Strong and early monkeypox virus-specific immunity associated with mild disease after intradermal clade-IIb-infection in CAST/EiJ-mice

Christian Meyer zu Natrup[1], Sabrina Clever[1], Lisa-Marie Schünemann[1], Tamara Tuchel[1], Sonja Ohrnberger[1] & Asisa Volz ®[1,2] ✉

Monkeypox virus (MPXV) is a zoonotic poxvirus long endemic in West and Central Africa. Outbreaks, first the global spread of clade II outside Africa in 2022, and since 2023 the accelerating spread of clade I in central Africa, point to MPXV adaptations that pose the risk of it becoming more transmissible in humans. Animal models mimicking the clinical disease outcome in humans are important to better understand pathogenesis, host tropism, and the contribution of genetic mutations. Here, we demonstrate that MPXV infection via tail scarification in CAST/EiJ mice is an appropriate animal model to mimic human mpox. In our study, disease outcome is milder in clade IIb than clade IIa-infected mice, which is associated with enhanced immunogenicity early during infection. This suggests that clade IIb more efficiently activates host immune responses, highlighting how this animal model could facilitate studying new MPXV variants to help develop efficient antivirals and preventive measures.

Mpox is caused by monkeypox virus (MPXV), a member of the genus orthopoxvirus, which in humans causes a systemic disease with a characteristic skin rash very reminiscent of human smallpox. Clinical manifestation in humans is mainly characterized by systemic disease with fever, headache, shortness of breath, swollen lymph nodes and skin lesions. Mpox has long been endemic in Africa as a zoonotic disease that can be transmitted to humans and other mammals from a rodent reservoir, followed by limited human-to-human transmission that is self-limiting. There are two distinguishable strains depending on the geographical distribution. Traditionally, the Central African strains (assigned as clade I MPXV) induced a severe and lethal disease with a case fatality rate (CFR) of up to 10%[1]. In contrast, the West African strains (clade II MPXV) caused milder clinical symptoms and a CFR of 2%.

Since 2022 an ongoing outbreak situation outside of Africa has resulted in more than 94000 confirmed human cases worldwide caused by a new MPXV clade II variant (now called clade IIb)[2]. Moreover, since the beginning of 2023, the alarmingly accelerated spread of the more severe clade I MPXV in the Democratic Republic of Congo (DRC) has now been declared a public health emergency of international concern by the WHO[3]. A new epidemiology feature of both outbreaks is sustained and efficient human-to-human transmission without any animal reservoir to initiate zoonotic MPXV infection. Close skin contact is hypothesized to be the main infection route for efficient human-to-human MPXV transmission. The global clade IIb mpox epidemic remains mild in immunocompetent hosts in terms of clinical disease, with a more localized skin rash at distinct sites on the body and low CFR, while the new clade Ib infections in Africa cause more severe and also lethal disease outcomes in humans[4,5].

Currently, little is known about the genetic basis accounting for the differences in MPXV virulence and host tropism, nor the underlying molecular mechanisms. MPXV adaptations pose the risk of MPXV

[1]Institute of Virology, University of Veterinary Medicine Hannover, Hanover, Germany. [2]German Centre for Infection Research, Partner Site Hannover-Braunschweig, Hannover, Germany. ✉e-mail: asisa.volz@tiho-hannover.de

becoming a more transmissible and maybe more virulent human pathogen with pandemic potential[4].

MPXV isolated from the 2022 outbreak was assigned as clade IIb based on genome sequencing and its high similarity to clade IIa MPXV[6–8]. The sequence differences detected between these clades pose the question of whether these changes contribute to the altered MPXV virulence and epidemiology seen in the international outbreak. Similar questions now arise for the alarmingly accelerated spread of the new MPXV clade Ib in Africa, where genomic sequence differences have also been demonstrated[9].

Animal models that mimic the clinical disease outcome in humans are important to better understand MPXV pathogenesis, and the possible contribution of new clade strain mutations. Different animal models for MPXV have been established, including non-human primates (NHP), prairie dogs, and different rodents. NHP and prairie dogs are highly susceptible to MPXV infection, resulting in a systemic viral infection and generalized pustular rash. However, NHP or prairie dog experiments are difficult to perform in terms of ethics, costs, and available reagents[10,11]. A very recent study established the peridomestic African rodent *Mastomys natalensis* as a new rodent model susceptible to mild to asymptomatic MPXV clade IIb infection using intraperitoneal, rectal, vaginal, and transdermal inoculation[12]. Among the rodent models, mice are of special interest due to the ease of experimental studies and range of laboratory reagents available. Classical inbreed mice are not susceptible to MPXV infection. However, as established by Americo et al.[13,14], CAST/EiJ mice are highly susceptible to MPXV infection via intranasal and intraperitoneal routes. Americo and coworkers demonstrated that the clade I strain results in a severe and lethal infection in these mice. Clade IIa infection also induced a systemic and severe disease outcome, while the very recent clade IIb appeared attenuated with only mild clinical symptoms.

Here, we focused on clade II MPXV and demonstrate that infection via scarification using a dose of $2 \times 10^5$ PFU induced a systemic disease and manifestation of characteristic skin lesions in CAST/EiJ mice. MPXV clade IIa intradermal infection induced a more pronounced disease than MPXV clade IIb infection, which was associated with an increased and earlier activation of MPXV-immune responses. Regarding the skin lesions and systemic disease outcome, the intradermal infection route in CAST/EiJ mice provides a suitable animal model to study the pathogenesis, virulence and efficacy of preventive, therapeutic and vaccination measures against MPXV.

## Results

### MPXV skin scarification induces a clinical disease outcome in CAST/EiJ mice

To comparatively evaluate the clinical disease manifestation of MPXV clade IIa and IIb in CAST/EiJ mice, we used two different MPXV isolates. For MPXV clade IIa, we used a virus isolated from a wild-living monkey, a sooty mangabey, found dead in Taï National Park, Côte d'Ivoire, in March 2012[15]. For MPXV clade IIb, we received the MPXV isolated from the first confirmed mpox patient in Germany in May 2022[16]. This isolate was confirmed to be the causative agent of the 2022 epidemic in Germany. Both viral isolates showed comparable replication in MA-104 cells when characterized in multiple-step and one-step viral growth analysis (Fig. 1). Furthermore, plaque assays verified the findings of Americo and colleagues, indicating that MPXV clade IIa tends to release a greater number of virus particles into the medium, as evidenced by the increased formation of satellite plaques extending from larger, round plaques (Supplementary Fig. 1).

Americo and coworkers confirmed that CAST/EiJ mice were highly susceptible to MPXV clade IIa infection when using the MPXV-USA-2003-044 isolate, whereas MPXV clade IIb intranasal infection did not result in clinical disease outcome. We repeated these studies by evaluating the effects of intranasal infection with $2 \times 10^5$ PFU of the MPXV

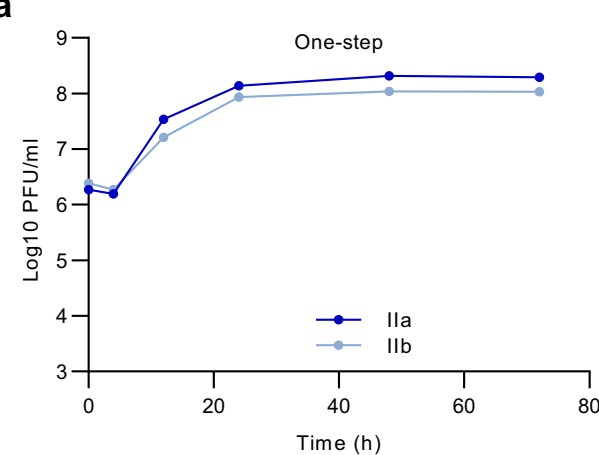

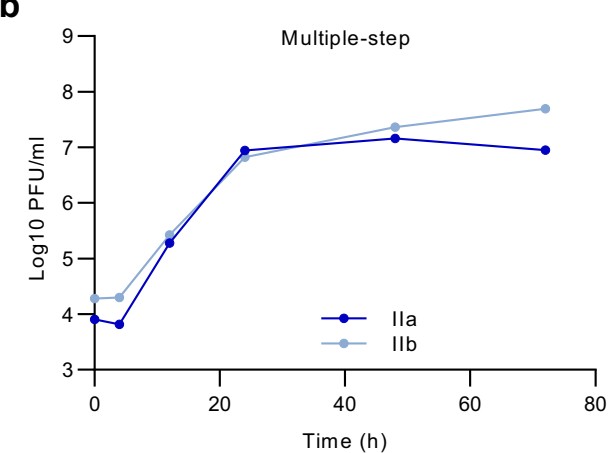

**Fig. 1 | Similiar in vitro replication of MPXV clade IIa and IIb in MA-104 cells.** MA-104 cells were infected with (**a**) MOI 3 or (**b**) MOI 0.05 and supernatants and cells were collected at the indicated time points. The viral titre (PFU/ml) was further determined for each time point using virus titration. Source data are provided as a Source Data file.

clade IIa isolate from 2012 and MPXV clade IIb isolate from 2022 (above) in CAST/EiJ mice (Fig. 2).

All mice were monitored daily for changes in body weight, mortality, and signs of mpox disease using a scoring system as described in the methods section (Table 1, Table 2). Starting 3 days post infection (dpi), all MPXV clade IIa infected mice demonstrated prolonged body weight loss (Fig. 2a) alongside signs of illness and mortality (Fig. 2b, c). Three mice died 5 dpi and another mouse had to be euthanized due to a humane endpoint assessment, including weight loss amounting to 21% of their initial body weight (Fig. 2a, b). In contrast, no body weight loss or systemic disease could be detected for the MPXV clade IIb infected mice. PBS infected mice served as control mice. No skin lesions were detected in the mice after intranasal infection (Fig. 2d). At the end of the experiment, we determined viral loads in the lung, liver, and spleen. Confirming previous results, we detected substantial MPXV titres in organs of MPXV clade IIa infected animals (Fig. 2e, f). In addition, substantial titres of MPXV were also measured in oropharyngeal swabs on day 4 and even higher titres on the day of death 5 or 6 dpi. No MPXV was detected in skin swabs (Fig. 2g, h). None of the clade IIb infected mice mounted titres of infectious MPXV in organs, while viral titres detected in the oropharyngeal swabs were significantly lower than the clade IIa infected animals. Again, no MPXV was measurable in the skin swabs of clade IIb infected mice (Fig. 2e–h).

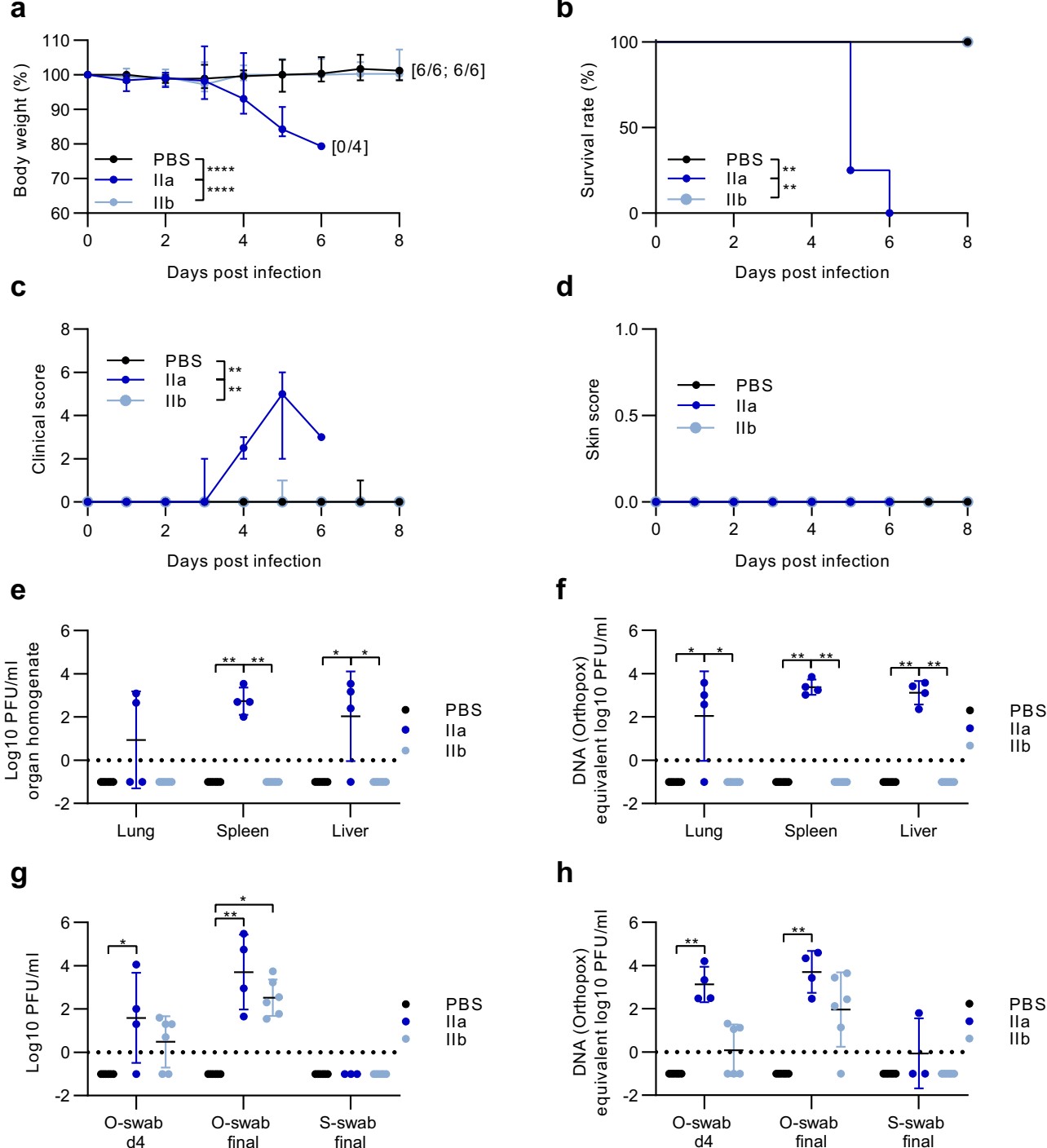

**Fig. 2 | Intranasal infection with MPXV clade IIb induces non-lethal disease in CAST/EiJ mice.** CAST/EiJ were intranasally infected with $2 \times 10^5$ PFU of MPXV clade IIa ($n = 4$), clade IIb ($n = 6$) or PBS ($n = 6$) as a control. Over the course of an 8-day infection period, daily monitoring of (**a**) body weight and (**b**) survival was conducted. Additionally, (**c**) general conditions and (**d**) the progression of skin lesions were assessed using clinical scores. Organs (lung, spleen, liver) were taken at the time point of death and analysed for (**e**) the amounts of infectious MPXV PFU/ml organ homogenate and (**f**) MPXV-DNA. Oropharyngeal swabs (o-swab) and skin swabs (s-swab) were taken on day 4 post-infection and on day of death (final), and analysed for (**g**) the amounts of infectious MPXV PFU/ml. **h** The amount of MPXV DNA was determined by qPCR assays. Titration and qPCR were performed once with two technical replicates. Error bars show the median ± 95% confidence interval (CI) (a, c, d) or the geometric mean ± geometric standard deviation (SD) (e-h). The area under the curve (AUC) was calculated for the continuous clinical data (a-d) and grouped AUCs were further used to analyse significant differences between groups over the entire infection period. *P* values were determined by one-way ANOVA with Tukey's multiple comparisons test (a) and by Kruskal-Wallis test with Dunn's multiple comparisons test (b–h). ****$p < 0.0001$ PBS versus IIa and IIa versus IIb in a. **$p = 0.0018$ PBS versus IIa and IIa versus IIb in b. **$p = 0.0082$ PBS versus IIa and IIa versus IIb in c. **$p = 0.0018$ PBS versus IIa (spleen) and IIa versus IIb (spleen), *$p = 0.0125$ PBS versus IIa (liver) and IIa versus IIb (liver) in e. *$p = 0.0125$ PBS versus IIa (lung) and IIa versus IIb (lung), **$p = 0.0019$ PBS versus IIa (spleen) and IIa versus IIb (spleen) and PBS versus IIa (liver) and IIa versus IIb (liver) in f. *$p = 0.0353$ PBS versus IIa (o-swab d4), **$p = 0.006$ PBS versus IIa (o-swab final), *$p = 0.0223$ PBS versus IIb (o-swab final) in g. **$p = 0.002$ PBS versus IIa (o-swab d4), **$p = 0.0038$ PBS versus IIa (o-swab final) in h. Dotted line: detection limit. Source data are provided as a Source Data file.

**Table 1 | Clinical Parameters used to score MPXV-infected CAST/EiJ mice**

| Parameter | Characteristics | Score |
|---|---|---|
| Social behavior/ General well-being/ Physical mobility | Normal conditions | 0 |
| | Ruffled fur, hunched back, complete mobility | 1 |
| | Ruffled fur, hunched back, restricted mobility | 3 |
| Lower respiratory tract | Normal conditions | 0 |
| | Low abdominal breathing | 1 |
| | Distinct abdominal breathing | 2 |
| | Labored breathing | 3 |
| | Labored breathing >6 h | 4 |
| | Pale oral mucosa | 4 |
| Upper respiratory tract | Normal conditions | 0 |
| | Mild serous eye or nasal discharge | 1 |
| | Moderate serous to purulent eye or nasal discharge | 2 |
| | Severe mucous to purulent eye or nasal discharge | 3 |
| | Severe mucous to purulent eye or nasal discharge with airways obstruction | 4 |
| Skin condition | Normal conditions | 0 |
| | Pustules or localized piloerection | 1 |
| | Visible scratches | 2 |
| | Papules or pustules and papules | 2 |
| | Active scratching | 2 |
| | Active scratching >12 h | 3 |
| | Extensive skin lesions | 3 |

**Table 2 | Parameters used to score skin lesions in MPXV-infected CAST/EiJ mice**

| Parameter | Characteristics | Score |
|---|---|---|
| Pox-specific skin lesions | Absence | 0 |
| | At the initial site of inoculation | 1 |
| | Disseminated skin rash | 2 |

To evaluate the outcome of intranasal MPXV infection in more detail, we also analysed the activation of different cytokines in the lung. Since type I interferons (IFN) are well-known to play important roles in anti-poxvirus defence, we measured the levels of IFN-α1, IFN-α2 and INF-β. We measured higher levels of all three in the MPXV clade IIb infected animals compared to the MPXV clade IIa infected mice (Supplementary Fig. 2). To evaluate the clinical outcome of MPXV intradermal infection, CAST/EiJ mice were infected with $2 \times 10^4$ or $2 \times 10^5$ PFU of either clade IIa or clade IIb MPXV by using skin scarification on the tail as a model for mpox transmission in humans. PBS mock-infected mice served as controls (Fig. 3). Starting 5 dpi all mice lost body weight, including PBS control mice, which exhibited a weight loss of about 7% compared to initial body weight. Importantly, no clinical disease symptoms were determined for the PBS-control mice despite the body weight loss. However, MPXV infection induced greater body weight loss, with a median of 10.2% for clade IIa MPXV-infected mice and a median of 10.5% for clade IIb-infected mice, averaged across both infection dosages on day 5. The pattern of body weight loss was comparable for both infection dosages, although somewhat more extreme for mice infected with clade IIb at the higher dosage (Fig. 3a, b).

Over the following days, PBS-infected control mice regained initial body weight by day 10. A similar pattern was observed for mice infected with $2 \times 10^4$ PFU of either MPXV clade until the end of the experiment on day 12 (median of 96.4% of initial body weight for MPXV clade IIa and median of 98.3% for MPXV clade IIb; Fig. 3a).

Mice infected with $2 \times 10^5$ PFU of either MPXV regained body weight by day 9 (median of 97.4% of initial body weight in clade IIa, median of 89.7% in clade IIb). However, unlike the $2 \times 10^4$ PFU infection dosage, all mice infected with $2 \times 10^5$ PFU again exhibited body weight loss starting on day 10. Here, MPXV clade IIa infected mice showed a

median of 13.7% body weight loss, and MPXV clade IIb infected mice had a median of 15.7% weight loss by day 12 (Fig. 3b).

No clinical disease or symptoms were observed in the PBS control group (Fig. 3e, f). After the $2 \times 10^4$ PFU infection, all MPXV clade IIb infected mice survived ($n = 8/8$), while only 62.5% of the clade IIa infected mice survived ($n = 5/8$; Fig. 3c). After the $2 \times 10^5$ PFU infection, 62.5% of the MPXV clade IIa mice ($n = 5/8$) and 42.9% of the MPXV clade IIb infected mice ($n = 3/7$) survived (Fig. 3d). These differences were not statistically different.

Following MPXV infection with either clade, all mice demonstrated clinical symptoms. The outcome of clinical disease was more pronounced in the $2 \times 10^5$ PFU challenge infection. Mice infected with $2 \times 10^4$ PFU of either MPXV clade showed mild clinical symptoms starting on day 5 when weight loss also started. These mice exhibited ruffled fur with the highest score of 1 between days 10–12 (Fig. 3e). For the $2 \times 10^5$ PFU challenge infection, the clinical disease outcome again started on day 5 and was significant with ruffled fur, hunched back, and respiratory symptoms (median of 1.5 for MPXV clade IIa and median of 1.0 for MPXV clade IIb). These clinical symptoms disappeared on day 9 but then returned with more severe outcomes on days 10 to 12 (maximal median of 4.0 for either MPXV clades). In addition to the symptoms seen before, the mice also showed subdued behaviour. For the $2 \times 10^5$ PFU challenge, the clinical disease outcomes were more severe in MPXV clade IIa infected mice (Fig. 3f).

Of note, all MPXV-infected mice also developed skin lesions at the initial site of infection on the tail starting on day 8–9 which progressed until the end of the experiment (Fig. 3g, h). For the $2 \times 10^4$ PFU challenge, skin lesions developed exclusively at the tail and were limited to 1 or 2 small lesions (median skin score of 1 for both MPXV clades; Fig. 3g). For the $2 \times 10^5$ PFU challenge, skin manifestations were more obvious and not only appeared on the tail but spread to other parts of the body. On day 11 pox lesions were also detected on the back around the spinal area and on the abdomen (median skin score of 2 for both MPXV clades; Fig. 3h–j).

## Systemic viral spread from the site of inoculation to internal organs late during infection

To evaluate the systemic viral spread from the primary infection site at the tail, we analysed MPXV titres in the liver, spleen, and lung by titration and qPCR at the day of death or the end of the experiment (12 dpi), and in oropharyngeal swabs taken on day 4 and at the day of death, and in a skin swab taken at the day of death (Fig. 4, Table 3).

For the $2 \times 10^4$ PFU infection study, we did not detect infectious MPXV for clade IIb in the liver and lungs, while two clade IIa infected animals mounted infectious virus in these organs (Fig. 4a, Table 3). For the spleen, we detected levels of infectious virus for MPXV clade IIa, but for clade IIb infection, only one infected animal mounted an infectious virus titre. Comparable viral titres in the different organs were analysed by qPCR (Fig. 4c). On day 4, we were unable to detect MPXV in oropharyngeal swabs except in one animal. On the day of death, oropharyngeal swabs revealed high titres of MPXV in clade IIa infected mice, in sharp contrast to very low titres in clade IIb infected mice (Fig. 4e, Table 3). On the skin, we detected similar viral titres for both clade IIa and IIb infected mice (Fig. 4e). These results were confirmed by qPCR (Fig. 4g).

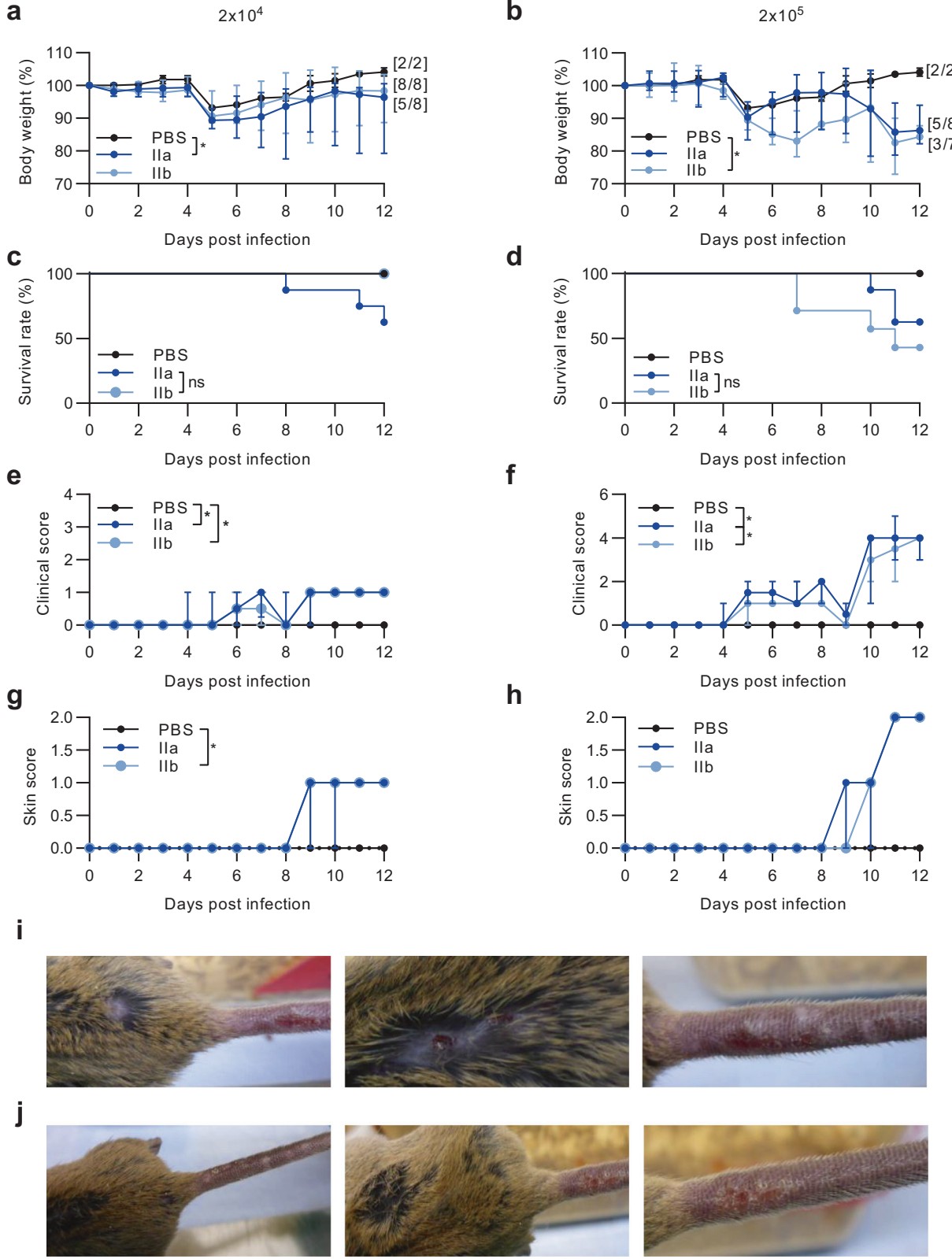

**Fig. 3 | Intradermal MPXV clade IIa infection induces a more severe clinical disease than intradermal MPXV clade IIb infection.** Animals were infected with a lower dose of $2 \times 10^4$ PFU or a higher dose of $2 \times 10^5$ PFU of MPXV clade IIa ($2 \times 10^4$: $n = 8$; $2 \times 10^5$: $n = 8$) or IIb ($2 \times 10^4$: $n = 8$; $2 \times 10^5$: $n = 7$) for 12 days using tail skin scarification. PBS served as controls ($n = 2$ each). During the infection period, daily monitoring of (**a, b**) body weight and (**c, d**) survival was performed. Additionally, clinical scores were used to assess (**e, f**) general conditions, and (**g, h**) the progression of mpox-specific skin lesions. The manifestation of skin lesions at different body sites were observed for $2 \times 10^5$ PFU MPXV clade IIa (**i**) and clade IIb (**j**) infected animals. Error bars show the median ± 95% confidence interval (CI). The area under the curve (AUC) was calculated and grouped AUCs were further used to analyse significant differences between groups over the entire infection period. $P$ values were determined by Kruskal-Wallis test with Dunn´s multiple comparisons test (b, c, d, g, h) and two-tailed Mann-Whitney test (a, e, f). *$p = 0.0444$ PBS versus IIa in a. *$p = 0.0379$ PBS versus IIb in b. *$p = 0.0444$ PBS versus IIa, *$p = 0.0222$ PBS versus IIb in e. *$p = 0.0444$ PBS versus IIa, *$p = 0.0281$ IIa versus IIb in f. *$p = 0.0176$ PBS versus IIb in g. ns = not significant. Source data are provided as a Source Data file.

For the $2 \times 10^5$ PFU challenge, we detected similar geometric mean titres of infectious virus in the liver of MPXV clade IIa and clade IIb infected mice (Fig. 4b, Table 3). Virus titres derived from the lungs and spleen were substantially higher for clade IIa compared to IIb-infected mice (Fig. 4b, Table 3). The pattern of MPXV titres in the organs was further confirmed by qPCR analysis (Fig. 4d). Again, we could not detect any infectious MPXV at 4 dpi in oropharyngeal swabs, but once more detected high titres from MPXV clade IIa and low titres from clade IIb infected mice on the day of death (Fig. 4f, Table 3). On the skin, we detected similar titres on MPXV clade IIa and clade IIb infected mice (Fig. 4f, Table 3).

### Early viral spread from the site of inoculation to internal organs at 8 dpi

Since results from the 12-day intradermal infection indicated that MPXV is systemically distributed to internal organs such as the liver and spleen, we aimed to evaluate MPXV-induced morbidity and mortality and viral spread earlier during infection and sacrificed the mice 8 dpi as established by Americo et al.[13,14]. We again infected CAST/EiJ mice intradermally with $2 \times 10^5$ PFU of clade IIa or clade IIb MPXV by skin scarification. PBS-infected mice served as controls. Again, the infected mice suffered from clinical disease and weight loss as described above. Briefly, mice exhibited weight loss at day 5 with a median of around 5% for both clades of MPXV (Fig. 5a). Of note, MPXV clade IIa infected mice started to regain weight on day 6 to reach about 98% of their initial body weight, but subsequently continued to lose weight until 8 dpi retaining a total of 96.9% of their initial body weight. On day 8, clade IIb infected mice regained body weight reaching 95% of their initial body weight (Fig. 5a). All animals survived the infection interval of 8 days (Fig. 5b). Again, clinical symptoms were comparable with those determined above, and included ruffled fur, hunched back, and elevated breathing (Fig. 5c). A small skin lesion was detected on the tail at the initial site of infection for individual mice. No spread of skin lesions to other regions was detected for either clade (Fig. 5d). No weight loss nor clinical symptoms were observed in the PBS-infected control mice. When we analysed infectious MPXV in the lung, spleen, and liver, we detected substantial viral titres in the organs for both MPXV clades (Fig. 5e, Table 3). The pattern of MPXV titres in these target organs was further confirmed by qPCR analysis (Fig. 5f).

Oropharyngeal swabs from the day of sacrifice (day 8) revealed that animals infected with clade IIa exhibited substantially higher titres of MPXV than clade IIb infected animals (Fig. 5g, Table 3). In 8 dpi skin swabs we detected substantial titres of infectious virus after both MPXV clade IIa and clade IIb infection (Fig. 5g, Table 3).

### Immune responses induced after primary intradermal MPXV infection

Since neutralizing antibodies (nAb) have been demonstrated to be essential for providing protection against mpox, we characterized the activation of MPXV-neutralizing antibodies at the end of the infection experiments with MPXV clade IIa or clade IIb (Fig. 6a–f) 8 and 12 dpi. We first used 50 PFU MPXV clade IIa as the virus to be neutralized in a plaque reduction neutralization test including 10% guinea pig serum as a source of complement (PRNT$_{50}$; Fig. 6a–c). For the $2 \times 10^4$ PFU infection study, we detected only marginal titres of nAb in serum from both clade infection groups at day 12, which is in line with the clinical disease outcome seen in this study (Fig. 6a). For mice infected with $2 \times 10^5$ PFU of either MPXV clade, we measured detectable titres of nAb on day 8 as well as day 12 (Fig. 6b, c). Counterintuitively, at day 8 we detected a geometric mean titre of 1:27 clade IIa nAb in MPXV clade IIb infected mice, whereas the MPXV clade IIa infection group did not mount a geometric mean titre beyond the detection limit using the PRNT protocol as described before. Similarly, neutralization of MPXV clade IIa virus was more obvious in serum from the mice that had been

infected with the MPXV clade IIb strain (Fig. 6b). At day 12 the titres of nAb further increased and were substantially higher in the MPXV clade IIb infected animals with a geometric mean of 1:58 vs. a geometric mean titre of 1:45 in the MPXV clade IIa infected animals (Fig. 6c).

We then evaluated serum-neutralizing titres against 50 PFU MPXV clade IIb virus with 10% guinea pig serum (Fig. 6d–f). In the $2 \times 10^4$ PFU infection, we detected same levels of MPXV clade IIb-neutralizing antibodies in serum from mice infected with MPXV clade IIa and MPXV clade IIb (geometric mean of 1:17 for both MPXV clades; Fig. 6d). Already at 8 dpi, MPXV clade IIb-neutralizing antibodies were apparently activated in the mice infected with clade IIb (geometric mean of 1:26), whereas titres were below the detection limit in the MPXV clade IIa infected mice (Fig. 6e). Low titres were also detected 12 dpi after the $2 \times 10^5$ PFU infection dosage (geometric mean of 1:14) for the MPXV clade IIa infected mice. In contrast, MPXV clade IIb infected mice mounted increased titres with a geometric mean of 1:62 PRNT (Fig. 6f).

To evaluate the outcome of immunogenicity in more detail, we also analysed the activation of different cytokines in the lung as major MPXV target organ. As established above, we measured the levels of IFN-α1, IFN-α2 and INF-β. Remarkably, higher levels of all three were detected in clade IIb infected mice compared to clade IIa infected mice. The difference between clade IIa and clade IIb infected mice was obvious in the lungs 8 dpi and further increased until 12 dpi (Fig. 6g–i). For proinflammatory cytokine IL6, we measured higher levels in the MPXV clade IIb infected animals compared to the MPXV clade IIa infected mice already 8 dpi. The IL6 levels further increased until 12 dpi (Supplementary Fig. 3a). In the clade IIb infected mice, TNFα, an inflammatory cytokine secreted during acute inflammation by macrophages/monocytes, was substantially activated 8 dpi and even more so 12 dpi compared to the clade IIa infected animals (Supplementary Fig. 3b). Levels of anti-inflammatory IL10 appeared to be comparable for the MPXV clade IIa and clade IIb infected animals (Supplementary Fig. 3c).

## Discussion

The 2022 global mpox outbreak resulted in unprecedentedly high numbers of human infections caused by MPXV clade IIb. Subsequent studies revealed substantial genome sequence variations in MPXV clade IIb isolates. However, whether this epidemic was enabled due to altered human behaviour or whether the genetic changes better adapted the virus to humans remains unclear. Presently, an mpox outbreak caused by a new clade I lineage has been declared a public health emergency in Africa; originally endemic in the Democratic Republic of Congo (DRC) it has now spread to many other central African countries and beyond. Again, the impact of the genomic variations for the outcome of pathogenesis is not known[9].

In vivo, studies analysing the correlation between genome variations and the virulence of different MPXV clades are very limited due to biosafety requirements and the availability of appropriate animal models. Since the skin manifestations seem to play an important role in current (global and African) mpox epidemiology in terms of the route of transmission and clinical symptoms[17–19], an animal model mimicking the skin rash would have huge advantages.

Using the intradermal route, we show here that CAST/EiJ mice are susceptible to both MPXV clade IIa and clade IIb infection. These results are in line with previous studies from Americo and coworkers showing that CAST/EiJ mice suffer from systemic mpox disease when inoculated using intranasal or intraperitoneal infection routes[13,14]. Since our MPXV clade IIa and clade IIb isolates had not been evaluated in vivo before, we first confirmed that intranasal infection with $2 \times 10^5$ PFU induces similar virulence in CAST/EiJ mice as reported by Americo and coworkers. In our study, all clade IIa mice suffered from severe systemic disease and all succumbed to infection within 6 days, whereas no disease outcome was observed for the intranasal clade IIb infection. Based on these data, we established a model mimicking the

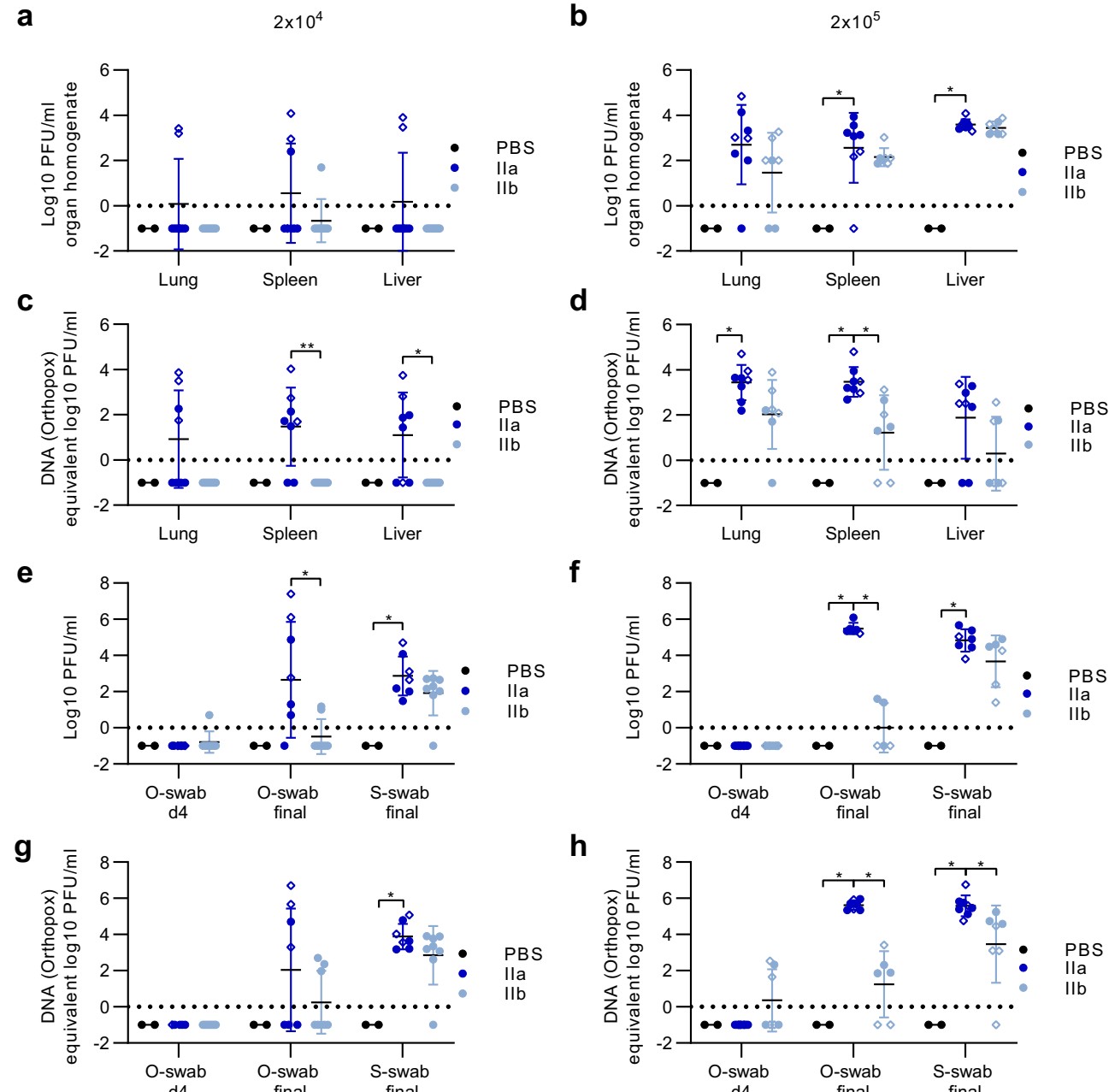

**Fig. 4 | Systemic spread of MPXV clade IIa and clade IIb after intradermal infection 12 days post infection.** CAST/EiJ were infected with $2 \times 10^4$ PFU or $2 \times 10^5$ PFU of MPXV clade IIa ($2 \times 10^4$: $n = 8$; $2 \times 10^5$: $n = 8$) or IIb ($2 \times 10^4$: $n = 8$; $2 \times 10^5$: $n = 7$) using tail skin scarification. PBS served as a control ($n = 2$ each). Mpox-specific target organs (lung, spleen, liver) were prepared at the day of death and assayed for (**a, b**) infectious virus titres and (**c, d**) viral DNA after $2 \times 10^4$ PFU and $2 \times 10^5$ PFU infection. In addition, oropharyngeal swabs (o-swab) and skin swabs (s-swab) were taken on day 4 post infection and on the day of death (final) and analysed for (**e, f**) the viral load (PFU/ml) and (**g, h**) the amount of viral DNA. Dot shapes (shaded blue as required) represent mice that were sacrificed at the end of the experiment on day 12 and rhombus shapes illustrate mice that were euthanized prematurely due to humane endpoint assessment. Titration and qPCR were performed once with two

technical replicates. Error bars show the geometric mean ± geometric standard deviation (SD). *P* values were determined by Kruskal-Wallis test with Dunn´s multiple comparisons test (a-d, f-h) and two-tailed Mann-Whitney test (e). *$p = 0.0295$ PBS versus IIa (spleen), *$p = 0.0482$ PBS versus IIa (liver) in b. **$p = 0.0078$ IIa versus IIb (spleen), *$p = 0.023$ IIa versus IIb (liver) in c. *$p = 0.0302$ PBS versus IIa (lung), *$p = 0.0202$ PBS versus IIa (spleen), *$p = 0.0158$ IIa versus IIb (spleen) in d. *$p = 0.0275$ IIa versus IIb (o-swab final), *$p = 0.0444$ PBS versus IIa (s-swab final) in e. *$p = 0.0458$ PBS versus IIa (o-swab final), *$p = 0.0234$ IIa versus IIb (o-swab final), *$p = 0.0319$ PBS versus IIa (s-swab final) in f. *$p = 0.0448$ PBS versus IIa (s-swab final) in g. *$p = 0.0225$ PBS versus IIa (o-swab final), *$p = 0.0167$ IIa versus IIb (o-swab final), *$p = 0.0142$ PBS versus IIa (s-swab final), *$p = 0.0192$ IIa versus IIb (s-swab final) in h. Dotted line: detection limit. Source data are provided as a Source Data file.

epidemiology and clinical symptoms observed in human mpox by a skin scarification infection route. Interestingly, we observed slight differences in the severity of the systemic disease between clade IIa and clade IIb infections.

Skin infection with $2 \times 10^5$ PFU of either MPXV clade in CAST/EiJ mice induced systemic infection with weight loss and a skin rash

closely resembling that of human mpox symptoms. Results from viral loads in the lung, liver, and spleen indicated that MPXV spreads systemically from the initial inoculation site at the tail to internal organs early during infection. The initial viremia is further supported by substantial weight loss in CAST/EiJ mice between days 4–8. A second viremia is hypothesized since we then detect skin lesions at different

**Table 3 | Summary of geometric mean MPXV titres in various organs**

| MPXV scarification | | | | | | | |
|---|---|---|---|---|---|---|---|
| | **12 dpi** | | | | **8 dpi** | | |
| **MPXV** | **Clade IIa** | | **Clade IIb** | | **Clade IIa** | **Clade IIb** | |
| Infection dosage | $2 \times 10^4$ | $2 \times 10^5$ | $2 \times 10^4$ | $2 \times 10^5$ | $2 \times 10^5$ | $2 \times 10^5$ | PFU |
| mice | | | | | | | |
| total | $n = 8$ | $n = 8$ | $n = 8$ | $n = 7$ | $n = 4$ | $n = 4$ | |
| survived | $n = 5$ | $n = 5$ | $n = 8$ | $n = 3$ | $n = 4$ | $n = 4$ | |
| lung | 1.2 | $5.0 \times 10^2$ | nd | 29.3 | $2.8 \times 10^2$ | 61.7 | PFU/ml |
| spleen | 3.6 | $3.7 \times 10^2$ | 0.2 (*) | $1.4 \times 10^2$ | $1.3 \times 10^3$ | $2.6 \times 10^2$ | PFU/ml |
| liver | 1.5 | $3.9 \times 10^3$ | nd | $2.8 \times 10^3$ | 8.4 | 5.0 | PFU/ml |
| o-swab | | | | | | | |
| day 4 | nd | nd | 0.2 (*) | nd | nd | nd | PFU/ml |
| final | $4.4 \times 10^2$ | $3.1 \times 10^5$ | 0.3 | 1.0 | $1.7 \times 10^4$ | 29.4 | PFU/ml |
| s-swab final | $7.4 \times 10^2$ | $6.8 \times 10^4$ | 83.6 | $4.7 \times 10^3$ | $4.2 \times 10^5$ | $5.3 \times 10^4$ | PFU/ml |

CAST/EiJ were infected with $2 \times 10^4$ or $2 \times 10^5$ PFU of either MPXV clade IIa or IIb for an infection interval of 12 days and with $2 \times 10^5$ PFU of either MPXV clade IIa or IIb for 8 days. Organs from day of necropsy and swab samples from day 4 and from day of necropsy (final) were collected and analysed for geometric mean infectious virus titres using plaque assays (data from Figs. 4 and 5). * Only one infected animal revealed infectious virus. nd = not detected.

sites on the body later during infection when mice once again suffered from weight loss between days 9–12. We detected substantial MPXV titres in these skin lesions. Based on these results, MPXV skin scarification in CAST/EiJ mice provides a suitable model to study MPXV pathogenesis.

Our findings from the skin infection model are consistent with the general pathogenesis of systemic orthopoxvirus (OPV) infection where, after primary replication at the site of infection, the respective OPV systemically spreads to internal organs such as the lung, liver, and spleen. Viremia is then followed by secondary viremia leading to an erythematous rash, as with variola virus (smallpox) infections in humans[20–23].

Scarification with a lower dosage of $2 \times 10^4$ PFU initially resulted in characteristic OPV pathogenesis as indicated by weight loss detected between days 4–8. However, skin lesions with the lower infection dosage were only detected locally around the tail as the initial site of replication. These findings suggest that effective viral clearance abrogates a second viremia from the internal organs to the skin. Viral clearance is further supported by the animals regaining body weight by 10 days post-infection (dpi). The MPXV organ titres further support this hypothesis: 12 dpi, MPXV titters were significantly lower or absent in the lung, spleen, and liver compared to the higher $2 \times 10^5$ PFU infection dosage, presumably due to efficient viral clearance after the lower infection dosage.

A dose response was also demonstrated in the intranasal (i.n.) infection studies by Americo. Here, $10^5$ PFU of MPXV clade IIa resulted in substantial weight loss, clinical symptoms, and substantial titres in lung, liver, and spleen. A 10-fold lower i.n. dose resulted in less pronounced body weight loss, milder clinical disease, and reduced viral titres in the target organs[13,14]. Reduced mortality and morbidity were also observed with $10^4$ PFU compared to $10^5$ PFU infection doses of MPXV clade IIa in the prairie dog model[24].

We also observed differences in virus replication for the $2 \times 10^5$ PFU infection dosage when comparing MPXV clade IIa and clade IIb: the former exhibited higher titres in the lung, spleen, and liver than MPXV clade IIb. However, both MPXV clades induced the manifestation of skin lesions starting 8 dpi. The appearance of skin lesions at different sites on the body indicates a second systemic viral spread from the internal organs via the bloodstream, supported by the MPXV titres detected in these organs, as mentioned above. The pattern of skin lesion development was very similar for all MPXV-infected mice in terms of the kinetics and body sites. Thus we consider a systemic viral spread to the skin more likely than auto-inoculation or transmission

from animal to animal. Auto-inoculation has been described for both CPXV and VACV infections in humans[25,26]. However, in these cases, the time kinetics and locations of the skin lesions differed. It will be interesting to further evaluate the impact of auto-inoculation and transmission following intradermal infection with MPXV clade IIa and clade IIb in the CAST/EiJ mouse model in future studies.

In addition, the slightly attenuated clinical outcome seen for the MPXV clade IIb infection is reflected by the lower MPXV titres in these target organs 12 dpi. Significantly, MPXV clade IIb titres were higher in lung and spleen when we analysed organ viral loads 8 dpi. These findings replicate those from the lower dosage infections above, indicating that MPXV clade IIb infection was cleared efficiently from the internal organs by the immune system. In direct contrast, following MPXV clade IIa infection, viral titres in these organs further increased from day 8 to day 12 post-infection.

The more efficient clade IIb viral clearance seems to be associated with a robust and early activation of MPXV-specific immune responses, as seen here for virus-neutralizing antibodies determined in the presence of complement and type I IFNs. Interestingly, mice infected with MPXV clade IIb mounted higher titres of neutralizing antibodies already at 8 dpi, and further increased at 12 dpi. Moreover, higher serum titres of MPXV-neutralizing antibodies were identified in MPXV clade IIb infected mice than in MPXV clade IIa infected mice, regardless of which virus was used in the neutralization assay. Thus, the more rapid and robust activation of neutralizing antibodies may contribute to the less severe disease outcome of clade IIb infections. These data further support that MPXV-neutralizing antibodies may also play an essential role in viral clearance.

These findings are in line with previous studies evaluating immunological correlates of vaccine-induced protection against mpox. Earl and coworkes demonstrated that the protective efficacy of MVA-and Dryvax-based vaccines against lethal intravenous MPXV infection in non-human primates (NHPs) is mainly based on neutralizing antibodies[27]. The essential role of MPXV-neutralizing antibodies is further supported by another study demonstrating that neutralizing antibodies are a correlate of the protective smallpox Dryvax vaccination in NHP after passive serum-transfer[28].

Type I IFN responses, as potent measures to protect against poxviral infections, were substantially increased in MPXV clade IIb infected mice. Interestingly, OPV have evolved multiple strategies to evade type I IFN responses[29,30]. For VACV and other OPVs, several immune evasion genes encoding proteins that specifically inhibit type I IFN signalling at different levels have been identified and characterised

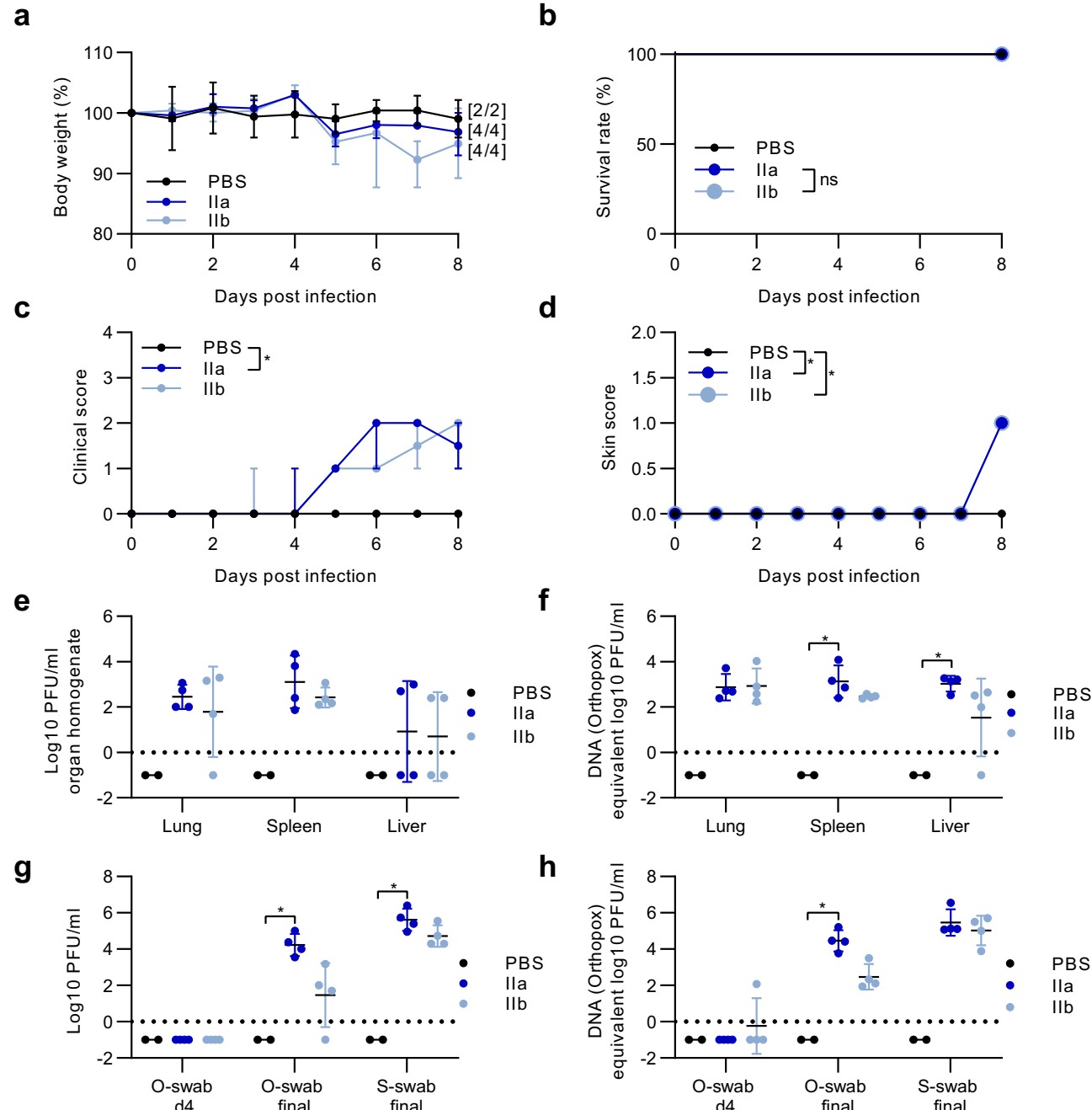

**Fig. 5 | Systemic spread of MPXV clade IIa and clade IIb after intradermal infection 8 days post infection.** CAST/EiJ mice were infected with 2x10⁵ PFU of MPXV clade IIa ($n = 4$) or IIb ($n = 4$) for 8 days using tail skin scarification. PBS served as a control ($n = 2$). Daily monitoring of (**a**) body weight and (**b**) survival was conducted, and (**c**) general conditions, or (**d**) the progression of skin lesions was assessed using clinical scores. Mpox-specific target organs (lung, spleen, liver) were harvested at the end of the experiment and assayed for (**e**) infectious virus and (**f**) viral DNA. Oropharyngeal swabs (o-swab) from 4 and 8 dpi (final) as well as skin swab samples (s-swab) from day 8 (final) were checked for (**g**) the amount of infectious virus (PFU/ml) and (**h**) orthopox DNA. Titration and qPCR were performed once with two technical replicates. Error bars show the median ± 95% confidence interval (CI) (a, c, d) or the geometric mean ± geometric standard deviation (SD) (e-h). The area under the curve (AUC) was calculated for the continuous clinical data (a–d) and grouped AUCs were further used to analyse significant differences between groups over the entire infection period. *P* values were determined by Kruskal-Wallis test with Dunn's multiple comparisons test. *$p = 0.0191$ PBS versus IIa in c. *$p = 0.0185$ PBS versus IIa and PBS versus IIb in d. *$p = 0.0336$ PBS versus IIa (spleen), *$p = 0.0475$ PBS versus IIa (liver) in f. *$p = 0.0363$ PBS versus IIa (o-swab final), *$p = 0.0379$ PBS versus IIa (s-swab final) in g. *$p = 0.0222$ PBS versus IIa (o-swab final) in h. Dotted line: detection limit. ns = not significant. Source data are provided as a Source Data file.

in vitro and in vivo[31–33]. Of note, some of these immune evasion genes have been also demonstrated to contribute to OPV host range[34].

From these findings, we hypothesize that some of the mutations acquired by MPXV clade IIb may favour the host sensing of MPXV by activating viral pathogen-associated molecular patterns. The loss or fragmentation of some of these genes in MPXV clade IIb might result in less pronounced inactivation of innate immune signalling as seen by the increased type I IFN responses, which will then support a strong activation of adaptive immune responses. This hypothesis is further supported by a recent study that identified changes in the orthologous poxvirus genes (OPG) *OPG153, OPG204,* and *OPG208* within the clade IIb genome that could be involved in the elevated activation of

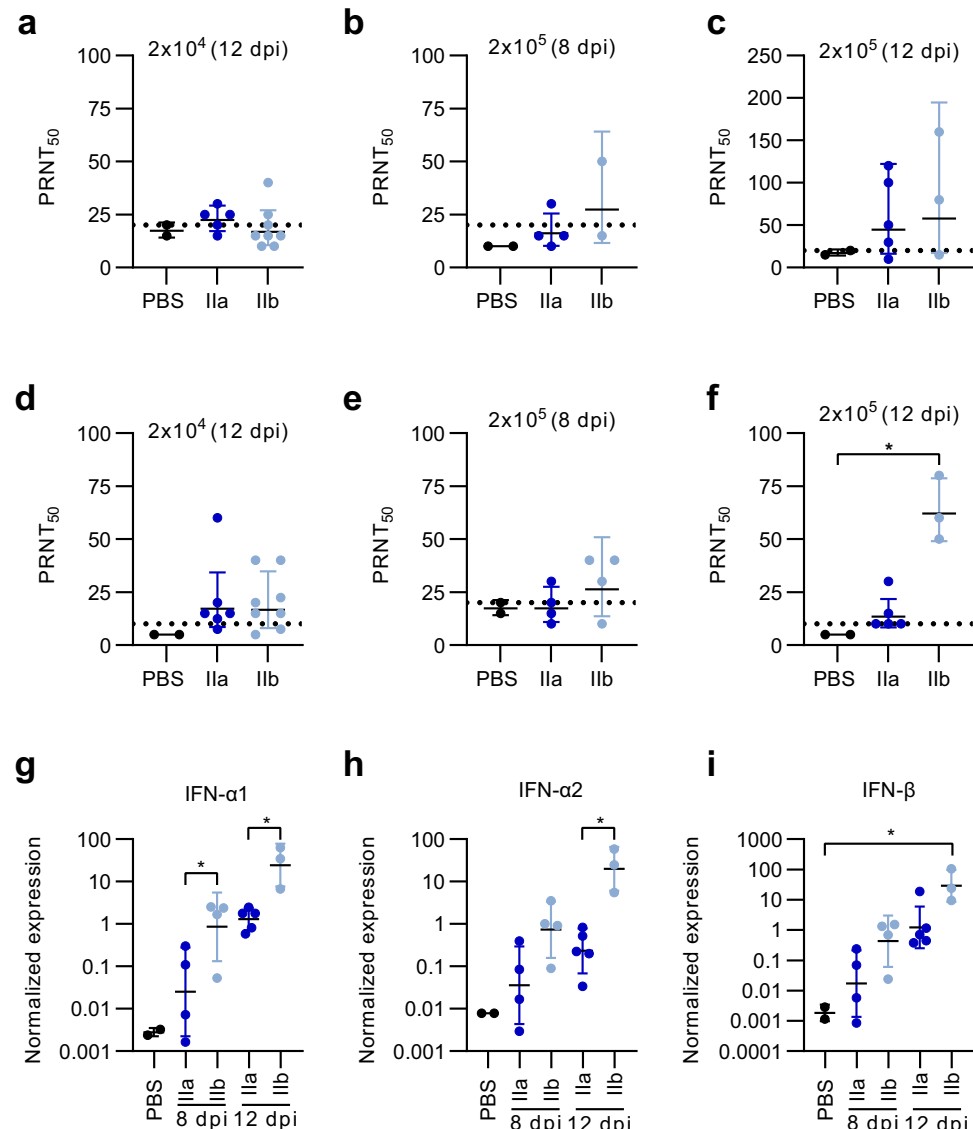

**Fig. 6 | Intradermal MPXV clade IIb infection activates stronger immune responses in CAST/EiJ mice.** After $2 \times 10^4$ PFU or $2 \times 10^5$ PFU MPXV infection, sera were collected on the day of necropsy (8 dpi or 12 dpi) and analysed for MPXV neutralizing antibodies against (**a**–**c**) clade IIa and (**d**–**f**) clade IIb MPXV. 50 PFU per well of MPXV virus were used for the complement-dependent antibody neutralization. mRNA was isolated from lung tissue to generate a cytokine profile, including IFN-α1, IFN-α2 and IFN-β. Cytokine levels are depicted as relative expression levels normalized against β-actin. Lung cytokine levels measured at day 8 and 12 post infection and analysed for (**g**) IFN-α1, (**h**) IFN-α2 and (**i**) IFN-β. Data from mice that were euthanized prematurely due to humane endpoint assessment are excluded. PRNT$_{50}$ and qPCR were performed once with two technical replicates. Error bars show the geometric mean ± geometric standard deviation (SD). *P* values were determined by Kruskal-Wallis test with Dunn´s multiple comparisons test (a-f) and one-way ANOVA with Dunnett's multiple comparisons test (g-i). *$p = 0.0176$ PBS versus IIb in f. *$p = 0.0395$ IIa versus IIb (8 dpi), *$p = 0.0351$ IIa versus IIb (12dpi) in g. *$p = 0.0475$ IIa versus IIb (12 dpi) in h. *$p = 0.0285$ PBS versus IIb 12 dpi in i. Dotted line: detection limit. Source data are provided as a Source Data file.

immune responses[35]. *OPG153* bears homology to VACV gene *A26L* encoding a major surface protein on the mature viral particle that is involved in the activation of antibodies, which might explain the increased activation of neutralizing antibodies in clade IIb infected mice. This is further supported by the comet assay results indicating an increased ratio of extracellular virus for the MPXV clade IIa[14]. However, since these results derive from very preliminary studies based on simple plaque formation assays, further more detailed studies evaluating the impact of intracellular versus extracellular viral particles on the activation of neutralizing antibodies are needed.

*OPG204* bears homology to VACV *B16R* coding for a secreted decoy receptor for type I IFNs, while *OPG208* has homology with VACV *B19R* encoding a potential virulence factor involved in inhibiting apoptosis in infected cells. In this context, several mutations in MPXV clade IIb have been correlated to APOBEC-3 activity[36]. Interestingly, an APOBEC-3-style mutation has been identified that disrupted the *A46R* gene[37]. *A46R* encodes a protein that specifically inhibits type I IFN signalling[38] which is further supported by our results. Another study identified that an MPXV clade IIb isolated from a patient in Germany harbours a translocation from the left end to the right end of the genome, which deleted *OPG005*[39]. *OPG005* is the homologue of the VACV *B22R/C16L* gene which codes for a protein that specifically inhibits IFN-β secretion by inhibiting IRF3 activation[40]. In summary, these data might explain the more pronounced activation of type I IFNs and the proinflammatory cytokine IL6 in the lung of the MPXV clade IIb group than the clade IIa group at day 8 and day 12. This is further supported by our results from the intranasal infection study where we did not detect viral loads in the lungs of the clade IIb group.

However, we did detect low titres of MPXV in oropharyngeal swabs of clade IIb-infected animals, which is indicative of MPXV clade IIb replication in the upper respiratory tract. This implies that MPXV clade IIb infection is blocked in the lung, resulting in the inability to spread further to internal organs, such as the liver and spleen. The increased activation of type I IFN responses as measured in the lungs of the clade IIb infection group might contribute to this block in viral replication and the subsequent outcome of clinical disease.

Such specific immunostimulatory properties associated with the loss of genomic information have already been seen for Modified vaccinia Ankara, MVA, which has lost about 30 kbp compared to the original vaccina virus, VACV. A similar evolution has been hypothesized for the variola virus (VARV) the causative agent of human smallpox. Compared to an ancestor virus, VARV has lost a range of genes involved in host range and immune evasion, which resulted in its restriction to the human host and increased virulence. These findings support the possibility that similar evolution has started for MPXV clade IIb, with increased adaption to humans. This might also be true for the MPXV clade Ib currently spreading in Africa. However, since little is known about the exact functions of these immune evasion genes, future studies are needed to correlate the observed genome sequence variations with their specific functions, something not specifically considered in this study. Future studies need to evaluate the contribution of selected immune evasion genes to the outcome of disease through generating specific mutant MPXVs. Here, the CAST/EiJ mouse model using the intradermal infection route could provide a valuable test system.

The mpox epidemic spreading in Africa has recorded increasing numbers of confirmed MPXV clade I infections in humans and a case fatality rate of 4.2%[41,42]. Again, the sustained human-to-human transmission stands out from the classical mpox epidemiology. Initial studies have identified novel sub-lineage clade Ib as causative agents of the epidemic. Following methods described here, the novel clade Ib could be investigated using the CAST/EiJ mice intradermal infection model to gain more insights into the characteristics of the strains causing this epidemic.

In conclusion, our study demonstrates that intradermal MPXV infection of CAST/EiJ mice provides a suitable animal model for human mpox. Importantly, the manifestation of skin lesions late in infection, as established for human mpox, also allows one to study the pathogenesis and transmission already observed in current epidemics. We also identified differences in the virulence of MPXV clade IIa and IIb, which will allow one to pinpoint genetic changes responsible for differences seen in epidemiology and clinical manifestations. The first indication that genes involved in MPXV immune evasion might drive these differences is that our MPXV clade IIb infected mice mounted higher titres of virus neutralizing antibodies early after infection. It would be highly informative to link the increased antibody activation to a specific gene function. In addition, the intradermal infection model with adequate manifestation of clinical mpox will also be advantageous for developing and evaluating antivirals, preventive measures, and candidate vaccines under more reproducible and reliable conditions.

## Methods

### Ethics statement
All animal trials were conducted in accordance with European and national regulations governing animal experimentation (European Directive 2010/63/EU and Germany's Animal Welfare Acts). These experiments were approved by the Niedersächsisches Landesamt für Verbraucherschutz und Lebensmittelsicherheit (LAVES) in Lower Saxony, Germany. The protocol was approved by the Commission according to section 15 of the German Animal Welfare Act and the Commission for Research Ethics of the University.

### Cell culture
MA-104 cells (ATCC® CRL-2378.1™) and Vero E6 cells (ATCC® CRL-1586™) were cultured in Dulbecco's Modified Eagle's Medium (DMEM) supplemented with 10% FBS, 1% non-essential amino acids (only for MA-104 cells), and 1% penicillin-streptomycin.

### Viruses
MPXV clade IIa (isolate West Africa/Taï National Park/ Côte d'Ivoire/ 2012), and MPXV clade IIb (isolate muc IMB-1/Germany/2022) were propagated on MA-104 cells in DMEM (Sigma-Aldrich) supplemented with 2% FBS, 1% non-essential amino acids, and 1% penicillin-streptomycin at 37 °C. MPXV stocks were amplified and purified using a modification of the protocols of Americo and Moss 2010: Monolayers of MA-104 cells were infected with 1 PFU/cell. After 3 days, infected cells and supernatants were harvested and lysed by three freeze-thaw cycles followed by extensive vortexing. Virus suspension was purified by centrifugation at $38,200 \times g$ for 90 min. Virus was extracted from the pellets by suspension in 1 mM Tris-HCl (pH 9), followed by extensive vortexing. Virus samples were divided into 200 µl aliquots, tittered and frozen at −80 °C. MPXV clade IIa stock revealed a titre of $2.8 \times 10^8$ PFU/ml, and MPXV clade IIb stock revealed a titre of $1.8 \times 10^9$ PFU/ml.

### Comet assay
MA-104 cells in 6-well plates were infected with diluted samples of MPXV. Infected cells were incubated for 1 h at 37 °C before medium was aspirated, and the infected MA-104 cells were washed with fresh medium to remove free virus. The plates were further incubated at 37 °C for 48 h, after which they were fixed and stained with crystal violet.

### Viral replication
MA-104 cells in 6 well plates were infected with an MOI 0.05 (multiple step growth curve) and MOI 3 (one step growth curve) of MPXV clade IIa and IIb. After incubation of 1 h at 37 °C, cells were multiply washed and further incubated in fresh DMEM supplemented with 2% FBS, 1% non-essential amino acids, and 1% penicillin-streptomycin. After 0, 4, 12, 24, 48 and 72 h, cells and the supernatant were collected and stored at −80 °C. After repeated freeze-thawing, the viral titre was determined for each time point using virus titration on Vero E6 cells: Briefly, Vero E6 cells were aliquoted in 6 well plates and virus suspension was added in serial 10-fold dilution steps. After 1 h, cells were washed and overlayed with medium for 48 h. Plates were then fixed in 4% paraformaldehyde/PBS and were further stained with a Vaccinia Virus (Lister Strain) rabbit polyclonal antibody (OriGene, cat.: BP1076, lot: 5G18723, 1:2000 diluted) and a secondary HRP-labelled goat anti-rabbit antibody (Biozol, cat.: JIM-111-035-144, lot: 166835, 1:5000 diluted). The signal was developed using a precipitate-forming TMB substrate (True Blue, KPL SeraCare, 5510-0030) and the number of plaque forming units (PFU) was counted.

### Experimental design
CAST/EiJ mice were initially obtained from Jackson Laboratories. Experiments were performed using animals of mixed sex (12–20 weeks old), provided from an in-house breeding colony maintained at the Research Center for Emerging Infections and Zoonoses (RIZ), University of Veterinary Medicine, Hanover, Germany. All animals ($n = 61$) were maintained under specified pathogen-free conditions and had free access to food and water. For challenge infection studies with MPXV, female and male CAST/EiJ mice were kept alone or in groups of two to five animals in individually ventilated cages (IVCs, Tecniplast). The animal trials are conducted under a 12 hour light/dark cycle, with the room maintaining a relative humidity ranging from 45% to 65% and a temperature between 20 °C and 24 °C. All work with infectious MPXV

in the animal stables and laboratories was performed in BSL-3 facilities at the RIZ, University of Veterinary Medicine, Hanover, Germany. The obtained study results were not categorised by sex because no sex differences were identified in CAST/EiJ mice.

In the first part of the study, mice ($n = 14$) were intranasally infected with $10\,\mu l$ containing $2 \times 10^5$ PFU MPXV clade IIa ($n = 4$, all males) or clade IIb ($n = 6$, 2 males and 4 females) or PBS ($n = 6$, 3 males and 3 females) as a control. The animals were harvested 8 dpi. In the second part, CAST/EiJ ($n = 35$) were intradermally infected for 12 days using tail skin scarification as described in Melamed et al.[43]. For this, a droplet of $10\,\mu l$ containing $2 \times 10^4$ or $2 \times 10^5$ PFU MPXV clade IIa ($2 \times 10^4$: $n = 8$, 6 males and 2 females; $2 \times 10^5$: $n = 8$, 5 males and 3 females) or IIb ($2 \times 10^4$: $n = 8$, 6 males and 2 females; $2 \times 10^5$: $n = 7$, 3 males and 4 females) or PBS (both infection dosages: $n = 2$, all males) was placed on the tail and rubbed into the skin using a cannula. In the last setup, this scarification methodology was repeated with a shorter infection period of 8 days ($n = 10$) using $2 \times 10^5$ PFU MPXV clade IIa ($n = 4$, all females) or IIb ($n = 4$, all females) or PBS ($n = 2$, all females). Following the challenge infection, mice were observed at least twice daily to assess their well-being, overall health, and clinical signs. Clinical scoring criteria were based on specific parameters such as social behaviour, general well-being, physical mobility, fur condition, upper and lower respiratory tract and skin condition. The cumulative clinical scores for each animal were determined at least once per day and used to evaluate the systemic mpox disease.

Skin score based on the distribution of pox-specific skin lesions in the infected animals was determined. The skin score system ranges from score 0 to 2. A skin score of 0 indicates the absence of pox-specific skin alterations, whereas a score of 1 reflects pox-specific skin alterations at the initial site of inoculation. A skin score of 2 evinces additional disseminated skin rash/pox-specific skin alterations beyond of the initial inoculation site. Body weights were checked daily. For MPXV challenge infection, animals were kept in individually ventilated cages (IVCs, Tecniplast) in approved BSL-3 facilities. All animal and laboratory work with infectious MPXV was performed in a BSL-3e laboratory and facilities at the Research Center for Emerging Infections and Zoonoses (RIZ), University of Veterinary Medicine, Hanover, Germany.

### Virus titration from organs and swab samples
Organ samples of infected mice were homogenized in 1 ml DMEM supplemented with penicillin and streptomycin (Gibco) and were stored at $-80\,°C$. Viral titres were further determined on VeroE6 cells. Briefly, organ homogenates and swab samples were added in duplicates to 96-well plates in serial 10-fold dilution steps. After 48 h at $37\,°C$, cells were fixed in 4% paraformaldehyde/ PBS and then stained as described above. The virus titre was calculated as PFU/ml organ homogenate.

### Quantification of viral DNA
Genomic DNA was extracted from collected lung, spleen, liver, and swab samples using the commercial NucleoMag®Tissue kit (MACHERY-NAGEL) following the manufacturer´s instructions. The orthopox DNA was amplified in a CFX96-Touch Real-Time PCR system (Bio-Rad) using the commercially available Luna® Universal SYBR-based KIT (NEB, M3003E) with 5´-TAATACTTCGATTGCTCATCCAGG-3´ as forward and 5´-ACTTCTCACAAATGGATTTGAAAATC-3´ as reverse primers. The primers target the *I7L* gene in the Vaccinia virus strain MVATGN33.1 Modified Virus Ankara (GenBank: EF675191.1) and are specified and established to detect orthopoxviruses[44]. The PCR program included 45 cycles of $95\,°C$ for 15 s (denaturation), and $60\,°C$ for 34 s (annealing and elongation). DNA dilutions were isolated from MPXV stock and used as PCR standards. The sample Ct value was compared to the standard to calculate the DNA quantity. Data collection and analysis were carried out with CFX Maestro 2.3 (Bio-Rad).

### PRNT$_{50}$
The protocol was adapted from Gilchuck et al.[45]. Serum samples were analysed for the quantity of neutralizing antibodies against MPXV clade IIa and clade IIb. For this, heat-inactivated serum was serially diluted 2-fold in duplicate in $50\,\mu l$ DMEM containing 2% FBS, 10% guinea pig complement and 1% penicillin-streptomycin. $50\,\mu l$ MPXV clade IIa or IIb (50 PFU) was further added and incubated for 2 h at $37\,°C$. The virus-serum suspension was then transferred onto Vero E6 cells and incubated again for 1 h at $37\,°C$. Lastly, each well was overlayed with $100\,\mu l$ of a 1:1 mixture of DMEM and Avicel RC-591 (Dupont, Nutrition & Biosciences) and 96-well plates were incubated for 48 h. Cells were then fixed in 4% paraformaldehyde/ PBS and immunostaining was performed as described above. Using the BioSpot Software Suite (CTL Switchboard 2.7.2, CTL Europe GmbH), the serum neutralization titre (PRNT$_{50}$) was determined as the reciprocal of the highest serum dilution that reduced plaque formation by more than 50%.

### Cytokine profile
RNA was extracted from $200\,\mu l$ of homogenized lung samples using the NucleoMag® RNA kit (MACHERY-NAGEL). Extracted mRNA was further amplified in a CFX96-Touch Real-Time PCR system (Bio-Rad) using the commercially available Luna® Universal One-Step RT-qPCR Kit (NEB, E3005). To analyse the cytokine profile, different primer pairs for IFN-α1 (forward: TAATTCCTACGTCTTTTCTTT, reverse: TATGCCTGATCCCTGAACAGT), IFN-α2 (forward: TTGAAGGTCCTGG-CACACAG, reverse: GAGGTTCAAGGTCTGCTGA), IFN-β (forward: AAGAGTTACACTGCCTTTGCCATC, reverse: CACTGTCTGCTGGTG-GAGTTCATC), IL6 (forward: CGGAGAGGAGACTTCACAGAG, reverse: CATTTCCACGATTTCCCAGA), IL10 (forward: TGCACTACCAAAGCCA-CAAG, reverse: TGATCCTCATGCCAGTCAGT) and TNFα (forward: AGCCAGGAGGGAGAACAGA, reverse: CAGTGAGTGAAAGGGACA-GAAC) were used. The expression levels were normalized against β-actin expression (forward: GTGGGCCGCTCTAGGCACCAA, reverse: CTCTTTGATGTCACGCACGATTTC). Each reaction started with an enzyme activation step at $95\,°C$ for 60 s, followed by 44 cycles consisting of denaturation at $95\,°C$ for 20 s and annealing and elongation at $56\,°C$ for 30 s. Data collection and analysis were carried out with CFX Maestro 2.3 (Bio-Rad).

### Statistical analysis
The clinical data (body weight, clinical score, skin score) are presented as individual values with the median ± 95% confidence interval (CI). Data of organ titration, qPCR and immune data are presented as individual values with the geometric mean ± geometric standard deviation (SD). The data were further tested for normal distribution using the Kolmogorov-Sminov test. The following tests were used for the non-normally distributed data: For comparisons involving three or more groups, the Kruskal-Wallis test with Dunn's multiple comparisons test was used, while the two-tailed Mann-Whitney test was employed for two-group comparisons. For normally distributed data, one-way ANOVA with Tukey's multiple comparisons test or Dunnett's multiple comparisons test was conducted to compare more than two groups. For the data of the body weight, scores and the survival rate, the individual values of the area under the curve (AUC) were determined and grouped AUCs were further used for calculation. Significant differences are thus related to the entire infection period. *P* value below 0.05 was considered statistically significant. All statistical analyses were carried out with GraphPad Prism 9.0.0. All the data generated in this study are provided in the Source Data file. Source data are provided with this paper.

### Reporting summary
Further information on research design is available in the Nature Portfolio Reporting Summary linked to this article.

## Data availability

All the data generated in this study are provided in the Source Data file. Source data are provided with this paper.

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

## Acknowledgements

This work was supported by the German Center for Infection Research (DZIF: projects TTU 01.944 to A.V.).

## Author contributions

A.V. conceptualized the study and acquired funding. A.V., C.M.Z.N., L.-M.S., and S.C. established the in vivo mouse model and performed in vitro and in vivo experiments. S.O. and T.T. assisted in the experiments. C.M.Z.N. and A.V. acquired data, interpreted data, and wrote and revised the manuscript together with all coauthors.

## Funding

## Competing interests

The authors declare no competing interests.
