## [Transparent Peer Review file · Nature Communications]

Strong and early monkeypox virus-specific immunity associated with mild disease after intradermal clade-IIb-infection in CAST/EiJ-mice

Corresponding Author: Professor Asisa Volz

Version 0:

Reviewer comments:

Reviewer #1

(Remarks to the Author)

Due to the ongoing global spread of the clade IIb mpox and increased cases of clade I mpox in central Africa, there is increased interest in poxvirus related research and developing research tools to address the emerging threat of mpox. Research tools needed are animal models (including small animal models) to study pathogenesis to better understand impact of mutations occurring from person-to-person spread as well as to evaluate vaccines and therapeutics.

Natrup, et al provide an early initial step to meet this goal by describing in great detail a cutaneous challenge model in CAST/EiJ mice, which are known to be susceptible to mpox via the intranasal and intraperitoneal routes. The key finding is that mice develop systemic disease after intradermal inoculation and have a disease course that might represent a primary and secondary viremia. Similar to intranasal and intraperitoneal route, Clade IIa is more virulent than clade IIb by intradermal infection. The basis for the virulence difference is not known.

The strength of the paper are the details provided about the model and the discussion section where the authors hypothesize that some of the mutations acquired clade IIb may favor the host sensing of MPXV which allow earlier innate and adaptive immune responses.

Major point to be addressed

Photos of skin lesions on tail and other parts of the body would enhance the manuscript. Also wonder if footpad challenge was tried since that may provide a more controlled route of inoculation.

Text and figure 6 legend should make it clearer that neutralization assays were done in the presence of complement and thus data represents complement-dependent antibody neutralization. Would also include in legend that assays were done with only 50 pfu of MPXV.

Minor points to be addressed

1. Line 41/42. Might be better said, The global clade IIb mpox epidemic remains mild in immunocompetent hosts in terms of clinical disease
2. Lines 91/92. Instead of "We supplemented," might be better said, we repeated
3. Line 97. Humane (not human)
4. Line 169. Would include why sac at day #8 was selected (a time before expected death and before skin lesions are seen).
5. Line 173. While dose of virus is indicated in Figure 5, would be helpful to include it in the text.
6. Line 189. Not day of death, day of sac.
7. Line 201. Are there really "substantial" neutralizing titers in at day 8 (figure 6b)? Seems that there may be a measurable value above background. May need to revise description of neutralization data given activity was to a small amount of virus and in the presence of complement.
8. Line 228. Would delete the second half of the sentence (which are likely responsible for the differences seen in this outbreak), since the next sentence better states the situation.

9. Line 366. What was the titer of stock of virus then used for mouse infections?

10. Line 389. ? source of CAST/Eij mice?

11. Line 403. What was the target gene used to quantify viral DNA?

12. Figure 2. Why wasn't Ilb included as a comparison? Analysis would have been strengthened by including Ilb to show these isolates gave similar results to those published by Americo, et al.

13. Figure 2e and 2f. Surprised that DNA levels in 2f were similar to PFU in 2e. Would have expected higher DNA levels than plaques.

14. Figure 3a and 3b. Why did PBS mice lose weight? Was something going on in the vivarium that was different than the time of intranasal challenge?

15. Figure 4. Since organ titers were taken on day of death or day of sac, is there a way to indicate by symbol color the mice that succumb to infection and those that reached the end of the experiment and were sacrificed at the experiment endpoint on day 12?

Fig. 6g to 6i. Any statistical analysis to support what is written in the text?

Reviewer #2

(Remarks to the Author)

The CAST mouse was previously shown to be highly susceptible to infection by MPXV via intranasal and intraperitoneal routes. The novel aspect of the current study is the demonstration that infection of CAST mice with MPXV by tail scratch causes skin lesions, transient weight loss and systemic spread. The Ilb strain appears more attenuated than Ilb by skin scratch but the difference is less than previously reported following intranasal or intraperitoneal infection making this model distinct. However, more work would be needed to explain the apparent greater immunogenicity of Ilb than Ilb as a factor in lower virulence.

General comments

Methods – Need to better describe how skin scores and clinical scores were determined. Were the skin scores based only on the site of inoculation or also on other sites on body? How were mice housed? Could the skin lesions on the body be due to auto-inoculation or neighboring mice rather than systemic spread?

Indicate in all panels the day or days when significance was determined.

Because of small numbers of animals and frequent outliers, geometric means might be better than arithmetic means in plots. Were infected cell lysates or purified MPXV used for infections? If lysates, were the titers similar? If the Ilb titers were lower, then more lysate containing viral proteins might have been administered which could increase antibody and cytokine responses.

Specific comments

Fig. 2. Since the study is a comparison of clade Ilb and Ilb, was the latter also tested by IN route? If yes then add.

Fig. 3 – Text says that clinical disease outcome was more severe for Ilb at high dose but difference seems slight. Difficult to interpret the significance inset – is the significance between PBS and Ilb and Ilb or also between Ilb and Ilb. On which day or days was significance determined?

Fig. 4 – It would be better to calculate geometric mean titers rather than arithmetic – would make difference between Ilb and Ilb in lung look less different due to outliers in Ilb.

Fig. 6. Why only 2 data points for Ilb in panel B and only 3 in panel C?. Difference in neut antibody appears greater at 12 than 8 days contrary to text that claims early response difference.

Version 1:

Reviewer comments:

Reviewer #1

(Remarks to the Author)

This is a revised manuscript. The authors provide an extensive response that adequately respond to reviewers' comments. They make the associated adjustments to the text.

Minor points to just further enhance the manuscript

Line 122/123. During the revisions, authors accidentally switched weight loss after scarification. Fig 3b show clade Ilb losing slightly more weight than clade Ilb.

Line 231. Word choice suggestion for this sentence. Would suggest "below" the detection limit (instead of "beyond" the detection limit)

Line 770. Typo. VACV (not VACB)

Since Supplementary Table 1 and Supplementary Table 2 just refer to a scoring system (and not the data collected), I think they should just be referred to once in the main text and then be included in the methods section.

Reviewer #2

(Remarks to the Author)

The authors have responded to my comments in a satisfactory manner. The most noteworthy result is that the authors provide a model for clade II skin infection of CAST mice that is superior to the IN and IP route of infection described previously.

Point-by-point response

We thank the reviewers for their insightful observations and comments, which have all been answered and have enabled us to resubmit an improved manuscript. Based on the reviewer's excellent suggestions and appropriate questions, we have now performed additional in vivo studies in our CAST/EiJ mouse model to evaluate the effects of intranasal infection with the MPXV clade IIb. Based on the Reviewer's comments and to be clearer about the main findings that MPXV clade IIb induced improved immunogenicity as a factor in lower virulence, we performed additional experiments to evaluate the innate immune responses induced by MPXV infection in CAST/EiJ mice in more detail. Additionally, because of the Reviewer's suggestion to re-evaluate the statistical analysis, we have made extensive modifications throughout the manuscript in the abstract, results, and discussion sections.

Below we list all reviewers' points in black and our answers in blue. Within the manuscript, edits in response to the reviewers' comments are highlighted in yellow.

Reviewer 1:

Major points:

1. Photos of skin lesions on tail and other parts of the body would enhance the manuscript. Also wonder if footpad challenge was tried since that may provide a more controlled route of inoculation.

This is a very appropriate suggestion, and we now include photos of the skin lesions in Figure 3i-j within the manuscript. The figure shows the MPXV-specific skin lesions at different body sites after intradermal infection with MPXV clade IIa and clade IIb.

To mimic the infection route mainly seen in the ongoing outbreaks of MPXV clade Ib and clade IIb, we decided to use an intradermal infection route, also considering that MPXV primarily spreads from person to person through close contact with lesions on the skin. To mimic this epidemiology, we also considered footpad injections to enable this infection route. However, due to animal protection laws, footpad injections are no longer permitted in Germany. This is even more valid since the CAST/EiJ mice are smaller than other wild-type inbred mice. Thus, we looked for alternatives that might mimic the epidemiology of the current outbreaks. Tschärke and Smith established the ear pinnae model as advantageous for the intradermal infection route compared to footpad injections¹. Again, since the CAST/EiJ mice are smaller, the ear pinnae is unsuited for intradermal infection with MPXV. Americo and coworkers also established scarification as an intradermal infection route in CAST/EiJ mice to evaluate the virulence and pathogenesis of MPXV^{2,3}. Americo established the intradermal infection by scarification on the shaved back. We modified this version of skin scarification since we wanted to avoid auto-inoculation that might occur via nicks or cuts. Therefore, we decided to use skin scarification via the tail, which does not require shaving, to mimic the intradermal infection to analyse systemic spread and the manifestations of skin lesions.

2. Text and figure 6 legend should make it clearer that neutralization assays were done in the presence of complement and thus data represents complement-dependent antibody neutralization. Would also include in legend that assays were done with only 50 pfu of MPXV.

This is a very useful suggestion. We have included this information in the text (lines 213 – 214, 226 - 227) and the figure legend 6 (lines 724 – 725).

Minor points:

1. Line 41/42. Might be better said, The global clade IIb mpox epidemic remains mild in immunocompetent hosts in terms of clinical disease.

This was a helpful suggestion. We have now changed it (line 40).

2. Lines 91/92. Instead of “We supplemented,” might be better said, we repeated

This was a good suggestion, and we have now exchanged the word (line 92).

3. Line 97. Humane (not human)

We are sorry to have missed this mistake and very much appreciate the attentiveness of the reviewer. The typos were corrected (lines 98, 703, and 729).

4. Line 169. Would include why sac at day #8 was selected (a time before expected death and before skin lesions are seen).

This is a very appropriate suggestion, and we have now included additional information on why we decided to use day 8 and day 12 for sacrifice.

Initially, we decided to use CAST/EiJ mice to establish the intradermal infection model for human mpox since Americo and Moss have identified this mouse strain as highly susceptible to MPXV infection, with deaths occurring between days 5 and 8 post-infection after 10^5 and 10^6 PFU dosage i.p. infection. For the 10^4 PFU infection dosage, morbidity and mortality were delayed a few days. Based on these data, we sacrificed mice at day 8 as an early time point to characterize intradermal MPXV infection. Here, we aimed to assess the systemic MPXV spread to the internal organs and the manifestation of clinical disease outcomes determined by the clinical score sheet (see Supplementary Tables). Importantly, this aligns with the general time kinetics of other orthopoxvirus infections and the general outcome of mpox disease in humans. Since MPXV infections in humans are characterized by a centrifugal skin rash that spreads to other parts of the body and starts within several days after the onset of the systemic disease symptoms, we initially decided for a later time point of sacrifice to characterize intradermal MPXV infection for clinical disease and skin manifestation. The day 12 time point is again further supported by the results from Americo and coworkers^{2,3} since they monitored the mice over 14 days. We have included this information within the text (lines 186-188) to clarify this.

5. Line 173. While dose of virus is indicated in Figure 5, would be helpful to include it in the text.

This is a very appropriate suggestion. We included this information now in line 189.

6. Line 189. Not day of death, day of sac.

Done. This was an appropriate suggestion, and we exchanged the word (line 205).

7. Line 201. Are there really “substantial” neutralizing titres in at day 8 (figure 6b)? Seems that there may be a measurable value above background. May need to revise description of neutralization data given activity was to a small amount of virus and in the presence of complement.

This is a valid question, and we fully understand the reviewer’s point. Based on the protocol from Gilchuck and coworkers, we established the PRNT assay with 50 PFU MPXV and 10% guinea pig complement⁴. The use of 50 PFU MPXV/50µl has been further supported by additional recently published studies⁵⁻⁷. In addition, Hubert and coworkers confirmed using 10% guinea pig serum as a source of complement for efficient MPXV neutralization in the 96-well plate approach. From this, we consider our PRNT results as valid. However, to better highlight the details of our MPXV-PRNT protocol, we rewrote it for clarity and also revised the description of neutralizing antibody titres appropriately (lines 210- 234).

8. Line 228. Would delete the second half of the sentence (which are likely responsible for the differences seen in this outbreak), since the next sentence better states the situation.

We thank the reviewer for the excellent suggestion and deleted the second half of the sentence (line 251).

9. Line 366. What was the titer of stock of virus then used for mouse infections?

We used 2×10^4 or 2×10^5 PFU of MPXV clade IIa or clade IIb stock for infection in the CAST/EiJ mice. MPXV clade IIa stock has a 2.8×10^8 PFU/ml titre, and MPXV clade IIb stock has a 1.8×10^9 PFU/ml titre. We diluted the stocks accordingly to infection dosages 2×10^5 PFU/animal or 2×10^4 PFU/animal and confirmed the correct titres by back titration of the infection inoculum used for animal experiments. We added this information in the Material and Method section (lines 427 – 434).

10. Line 389. ? source of CAST/EiJ mice?

The CAST/EiJ mice were obtained from Jackson Laboratories (The Jackson Laboratory 600 Main Street Bar Harbor, ME USA 04609). We performed in-house breeding for the experiments and used the female and male animals (lines 452 - 455).

11. Line 403. What was the target gene use to quantify viral DNA?

The primer pair is based on the orthopoxvirus quantitative PCR (OPV-qPCR) established by Schroeder and coworkers⁸. Primers bind to the orthopoxvirus gene I7L, which is also present in MPXV clade IIa and MPXV clade IIb. This OPV-qPCR has been previously established to detect different orthopoxviruses. We included this information in the Material and Method section (lines 493 – 495).

12. Figure 2. Why wasn’t IIb included as a comparison? Analysis would have been strengthened by including IIb to show these isolates gave similar results to those published by Americo, et al.

This is a very appropriate suggestion. We performed additional experiments to characterize MPXV clade IIb after intranasal infection. In line with results from Americo and coworkers^{2,3}, we confirmed the attenuation and milder clinical disease outcome of the MPXV clade IIb intranasal infection in CAST mice. We included this data within Figure 2 and the results section.

13. Figure 2e and 2f. Surprised that DNA levels in 2f were similar to PFU in 2e. Would have expected higher DNA levels than plaques.

This is a valid question, and we fully understand the reviewer's point. As described above, we used the OPV-qPCR established by Schroeder and coworkers. This PCR assay targets the gene I7L, which encodes for a cysteine protease that is synthesized late in the orthopoxvirus replication cycle. Previous studies demonstrated that the I7 cysteine protease is essential for viral replication since a conditional lethal mutant has been mapped to this locus⁹. In addition, Byrd and coworkers demonstrated that I7 protein is the main viral core protein proteinase required for virus core protein maturation. Based on these functions, we hypothesize that detection of the I7L gene by PCR is also indicative of active viral replication, and for this, the DNA levels are similar to the values of infectious viruses measured by titration in PFU. To our knowledge, no direct comparison between the Pan-OPV PCR and PFU titration has been published. We repeated the PCR and confirmed the results. This is further confirmed by the results from the newly performed intranasal MPXV clade IIb infection in the CAST/EiJ mice.

14. Figure 3a and 3b. Why did PBS mice lose weight? Was something going on in the vivarium that was different than the time of intranasal challenge?

This is a valid question, and we fully understand the reviewer's point. This cage of PBS control animals consisted of two male mice. Based on our detailed protocols from the anesthesia, the intradermal infection, and daily monitoring using the clinical score sheets as described in Material and Methods, nothing was different compared to the other infected mice and the intranasal control mice. So, based on these results, we conclude that nothing significant was going on in the vivarium that could have impacted the outcome of the MPXV infections. This is further supported by the fact that the PBS control mice that showed body weight loss did not show clinical disease symptoms or mounted viral loads. To clarify this, we rewrote the section in the results (lines 118 - 121).

15. Figure 4. Since organ titers were taken on day of death or day of sac, is there a way to indicate by symbol color the mice that succumb to infection and those that reached the end of the experiment and were sac'ed at the experiment endpoint on day 12?

This is an excellent suggestion. To differentiate between animals that succumb to infection and animals that survive until the end of the experiment, we now use different symbol shapes. Dot shapes represent mice sacrificed at the end of the experiment, and rhombus shapes illustrate mice euthanized prematurely due to humane endpoints. We also now explain this in the figure legend (lines 701 – 703). We did not change the symbol color to be able to clearly assign the data point to MPXV clade IIa (dark blue) or clade IIb (light blue) infection.

16. Fig. 6g to 6i. Any statistical analysis to support what is written in the text?

This is a very appropriate suggestion. Based on Reviewer's suggestion to re-evaluate the statistical analysis, the new statistical analysis indicated that the differences for cytokines IL-6, IL-10 and TNF- α between the groups are not statistically different. For this we decided to switch these figures within the Supplementary Information. Additionally, we analysed type I Interferon responses in the lungs, since these are well-known to play important roles in anti-poxvirus defence. The additional data are now presented in Fig. 6g-i. These data clearly indicate the higher cytokine expression in the lungs of the MPXV IIb infected animals which are statistically significant.

Reviewer 2:

However, more work would be needed to explain the apparent greater immunogenicity of IIb than IIa as a factor in lower virulence.

This is a very appropriate suggestion. We agree with the reviewer that more work would be needed to precisely explain the improved immune responses observed after MPXV clade IIb infection as a factor for the lower virulence. Based on the reviewer's suggestion, we performed additional experiments to better characterize and evaluate the improved immunogenicity of clade IIb infection in the CAST/EiJ mouse model. The innate immune responses are significantly involved in properly sensing viral pathogens and subsequent activation of immune signaling, culminating in the activation of adaptive immune responses. In this context, Type I interferons (IFN) have long been identified as key contributors to effective antiviral responses against poxviral infections¹⁰. For this, we characterized the type I IFN responses as measured by IFN- α and IFN- β in qRT-PCR assays. IFN- α and IFN- β were significantly increased in MPXV clade IIb infected mice compared to MPXV clade IIa infected mice in the intradermal infection model. This is further confirmed in the intranasal infection model. The increased activation of type I IFN probably contributes to the lower virulence of MPXV clade IIb infection. The increased activation of type I IFN could result from genetic differences between clade IIa and clade IIb, resulting in different immune evasion genes. Here, the hypothesis is that MPXV clade IIb lost some accessory genes involved in host interaction and immune evasion. In this regard, Vaccinia virus (VACV) and other orthopoxviruses have evolved specific strategies to escape type I IFN responses. These strategies include the synthesis of viral immune modulators that specifically inhibit IFN- α and IFN- β . Several VACV genes code for such specific inhibitors of IFN- α and IFN- β . One gene candidate is the VACV B22R/C16L gene, which has been demonstrated to inhibit IRF-3 signalling, resulting in the block of IFN- β secretion. Interestingly, a recent study demonstrated that an isolate of clade IIb MPXV harbors a translocation from the right end of the genome to the left end, which results in the deletion of the C16L gene, which encodes a protein responsible for the inhibition of IRF-3, and subsequently IFN- β ¹¹. This might explain the increased activation of IFN- β in MPXV clade IIb infected mice. Moreover, a recent study demonstrates that the new MPXV clade IIb suffers from substantial genomic differences defined by APOBEC-3 style mutations. These APOBEC-3-driven mutations disrupt the A46R gene, which encodes a protein with a Bcl-2 domain and inhibits the TLR and IL-1 β -mediated activation of Nf- κ B¹². This also results in the inhibition of type-I IFNs. Thus, clade IIb MPXV with the disrupted A46R will not be able to inhibit type-I IFNs so potently as clade IIa MPXV with a functional A46R. This might be an explanation for the increased immunogenicity. Another study further characterized the pattern of MPXV clade IIb genomic differences¹³. This study confirmed the presence of genomic accordions as a well-established mechanism of poxvirus adaptation. The three genes B19R, C12L, and A26L have been identified with patterns of genomic accordions and thus might contribute to the different disease manifestations and epidemiology, as seen in the clade IIb outbreak. B19R encodes an IFN-type decoy receptor that inhibits type I IFNs. C12L encodes a protein that inhibits apoptosis

and harbors a serin protease inhibitor motif. An exciting result from this study is that a gene located in a more conserved genome region harbors genomic accordions. This is the A26L gene involved in morphogenesis, and its inactivation results in decreased particle-to-PFU ratios, as seen by increased mature virion (MV) particles and decreased enveloped virion (EV) particles. Moreover, the A26 protein is the primary target of host antibody responses, which activates neutralizing antibodies. As shown by Americo and coworkers³, we confirmed the increased ratio of MV particles to EV particles for the MPXV clade IIb when we compared the comet formation as seen by the release of virus particles as small satellite plaques from larger round plaques. We hypothesize that the increased presentation of MV particles with MV surface proteins, including A26 as an essential protein for activating neutralizing antibodies, might be the explanation.

In summary, we hypothesize that the genetic differences identified by several studies might contribute to increased immunogenicity and milder disease outcomes, as seen by the MPXV clade IIb infection. However, to identify specific immunological mechanisms, extensive future work in the MPXV-CAST/EiJ mouse model, including generating mutant viruses for functional genetic studies, would indeed need to be done. However, this would be well beyond the scope of this current study, so to highlight these limitations, we revised this section of the discussion and included these different aspects and the references. We hope we have made this much clearer now.

General comments

1. Methods – Need to better describe how skin scores and clinical scores were determined. Were the skin scores based only on the site of inoculation or also on other sites on body?

This is a very appropriate suggestion, and we now include detailed information on clinical scoring systems within the manuscript in the material and methods section and the Supplementary Material in Table 1 and Table 2. We differentiate between the manifestation of systemic mpox disease and the manifestation of skin lesions. We included the scoring system in the Material and Method section (lines 468 - 476) and the Supplementary Information for both clinical manifestations. Supplementary Table 1 includes the general score sheet. Supplementary Table 2 consists of the score sheet used to assess the manifestation of skin lesions, whether these were absent, present, or at which body site. To further illustrate and highlight the manifestation of skin lesions, we also included photos of the skin lesions at the different body sites in figure 2.

2. How were mice housed? Could the skin lesions on the body be due to auto-inoculation or neighboring mice rather than systemic spread?

This is a very appropriate suggestion, and we agree with the reviewer that skin lesions due to auto-inoculation cannot be 100% excluded. In addition, transmission from animal to animal cannot be 100% excluded in cages that house several mice.

However, because we also observed the development of lesions in mice housed alone in cages, we conclude that this is due to the systemic spread in the second viremia. The mice were housed in individually ventilated cages (IVC) in groups of 3-4 mice, or in the case of incompatibilities with conspecifics, in individual husbandry. The following table shows the constellation of the housed animals and the maximal skin score per animal (Score 0 = no skin lesions, Score 1 = skin lesions at the initial infection site, Score 2 = disseminated skin rash). The animals marked with an “X” have been housed alone in the cages and

developed a disseminated skin rash (= Score 2) in the same timely manner as mice that were housed with several other mice. In these cases, it indicates that the skin lesions on different parts of the body are not due to neighboring mice.

animal ID	sex	infection	MPXV clade	dosage	infection period	cage number	maximal skin score	
1	f	scar	IIA	10 ⁴	12	1	1	
2	f	scar	IIA	10 ⁴	12		1	
3	m	scar	IIA	10 ⁴	12	2	1	
4	m	scar	IIA	10 ⁴	12	3	0	
5	m	scar	IIA	10 ⁴	12		1	
6	m	scar	IIA	10 ⁴	12		1	
7	m	scar	IIA	10 ⁴	12	4	1	
8	m	scar	IIA	10 ⁴	12		1	
9	m	scar	IIB	10 ⁴	12	5	1	
10	f	scar	IIB	10 ⁴	12	6	1	
11	f	scar	IIB	10 ⁴	12		1	
12	m	scar	IIB	10 ⁴	12	7	1	
13	m	scar	IIB	10 ⁴	12	8	1	
14	m	scar	IIB	10 ⁴	12		1	
15	m	scar	IIB	10 ⁴	12	9	1	
16	m	scar	IIB	10 ⁴	12		1	
17	m	scar	PBS		12	10	0	
18	m	scar	PBS		12		0	
19	f	scar	PBS		8	11	0	
20	f	scar	PBS		8		0	
21	f	scar	IIA	10 ⁵	8	12	1	
22	f	scar	IIA	10 ⁵	8		1	
23	f	scar	IIA	10 ⁵	8		1	
24	f	scar	IIA	10 ⁵	8		1	
25	f	scar	IIB	10 ⁵	8	13	1	
26	f	scar	IIB	10 ⁵	8		1	
27	f	scar	IIB	10 ⁵	8		1	
28	f	scar	IIB	10 ⁵	8		1	
29	m	scar	IIA	10 ⁵	12	14	2	x
30	m	scar	IIA	10 ⁵	12	15	2	x
31	m	scar	IIA	10 ⁵	12	16	2	x
32	m	scar	IIA	10 ⁵	12	17	2	x
33	m	scar	IIA	10 ⁵	12	18	2	x
34	f	scar	IIA	10 ⁵	12	19	2	x
35	f	scar	IIA	10 ⁵	12	20	0	
36	f	scar	IIA	10 ⁵	12		2	
37	m	scar	IIB	10 ⁵	12	21	0	
38	m	scar	IIB	10 ⁵	12		0	
39	m	scar	IIB	10 ⁵	12	22	2	x
40	f	scar	IIB	10 ⁵	12	23	2	
41	f	scar	IIB	10 ⁵	12		2	
42	f	scar	IIB	10 ⁵	12	24	2	
43	f	scar	IIB	10 ⁵	12		1	
44	m	scar	PBS		12	25	0	
45	m	scar	PBS		12		0	
46	m	in	IIA	10 ⁵	8	26	0	
47	m	in	IIA	10 ⁵	8	27	0	
48	m	in	IIA	10 ⁵	8		0	
49	m	in	IIA	10 ⁵	8		0	
50	m	in	PBS		8	28	0	
51	m	in	PBS		8		0	
52	m	in	PBS		8	29	0	
53	f	in	PBS		8	30	0	
54	f	in	PBS		8		0	
56	f	in	PBS		8		0	
57	f	in	IIB	10 ⁵	8	31	0	
58	m	in	IIB	10 ⁵	8	32	0	
59	m	in	IIB	10 ⁵	8	33	0	
60	f	in	IIB	10 ⁵	8	34	0	
61	f	in	IIB	10 ⁵	8		0	
62	f	in	IIB	10 ⁵	8		0	

This is further supported by the pattern and time kinetics of the skin lesions. For all MPXV-infected mice, we observed the first skin lesions about day 8 post-infection. Over the following days, additional lesions appeared at different body sites. We have now included pictures of these lesions in the manuscript (Figure 2i, j). Importantly, the dissemination and time point of the lesion's appearance at the tail and other body sites were comparable for all MPXV-infected mice, independently of the number of mice included within one cage. If this had been due to autoinoculation and/or transmission from one animal to the other, the appearance of lesions would have been delayed and not so regulated in the same timely manner. In addition, due to autoinoculation or transmission, one would expect to see more

lesions in the face region, since mice are known for self-grooming and interacting socially with other mice. For other orthopoviruses infections, autoinoculation and contact inoculation have been described^{14,15}. However, in these cases, the time kinetics were different and not ordered as we have observed in our experiments. Finally, to minimize the risk of autoinoculation and/or transmission, we decided to use skin scarification for the intradermal infection route since the tail does not require shaving of the animal's fur. As mentioned above, if we had used another body site for skin scarification infection, e.g. the back, as established by Americo and Moss², this would have required shaving. Shaving increases the risk of autoinoculation and transmission via skin nicks or cuts from one animal to the other. However, we agree with the reviewer that the impact of autoinoculation and transmission from mouse to mouse after intradermal infections of the different MPXV clades should be evaluated in future experiments. This will be extremely interesting. To clarify this, we included a section within the discussion (line 308- 315).

3. Indicate in all panels the day or days when significance was determined.

This is an appropriate suggestion and we supplemented this information in the figure legend. For the clinical data (body weight, clinical score, skin score, survival rate), the area under the curve (AUC) was determined for each mouse and the grouped AUCs (group IIa, group IIb, group PBS) were further compared using the appropriate statistical test. Like this, the calculated significance is related to the whole infection period. For the organ data, the day of sacrifice or day 4 (for oro swab d4 samples) is the day when significance was determined. For immune data (Figure 6), the days are already indicated in each panel.

4. Because of small numbers of animals and frequent outliers, geometric means might be better than arithmetic means in plots.

This is an appropriate suggestion. For titration, OPV-qPCR and immune data, we now use the geometric means \pm geometric standard deviation (SD). For clinical data over a time period (score and body weight), we now use the median \pm 95% confidence interval (CI) for the median to better consider the small numbers of animals and outliers.

5. Were infected cell lysates or purified MPXV used for infections? If lysates, were the titers similar? If the IIb titers were lower, then more lysate containing viral proteins might have been administered which could increase antibody and cytokine responses.

This is a very appropriate question, and we have now included additional information on preparing MPXV stocks. For infection, we used purified MPXV stocks. Viruses were purified using a modification of the protocols of Americo and Moss^{2,3}: Monolayers of MA-104 cells were infected with 1 PFU/cell. After 3 days, infected cells and supernatants were harvested and lysed by three freeze-thaw cycles followed by extensive vortexing. Virus suspensions were purified by centrifugation at 16000 rpm for 90 min. Virus was extracted from the pellets by suspension in 10 mM Tris-HCl (pH 9), followed by extensive vortexing. Virus samples were divided into 200 μ l aliquots and frozen at -80°C. Titration was performed as described in Material and Methods. The titre of MPXV clade IIb stock (1.8×10^9 PFU/ml) exceeded the titre of MPXV clade IIa stock (2.8×10^8 PFU/ml). We have now included this information in the Material and Method section (lines 427 - 434).

Specific comments:

1. Fig. 2. Since the study is a comparison of clade IIa and IIb, was the latter also tested by IN route? If yes then add.

This is an interesting point. As already mentioned for reviewer 1, we performed additional animal experiments to evaluate the intranasal infection with MPXV clade IIb. In line with results from Americo and coworkers^{2,3}, we confirmed the attenuation of the MPXV clade IIb intranasal infection in CAST mice. We included the data in Figure 2.

2. Fig. 3 –Text says that clinical disease outcome was more severe for IIa at high dose but difference seems slight. Difficult to interpret the significance inset – is the significance between PBS and IIa and IIb or also between IIa and IIb. On which day or days was significance determined?

This is a valid question, and we fully understand the reviewer's point. Regarding the clinical symptoms at high doses (clinical score, panel f), the significance is between PBS and IIa and IIa and IIb. The significance was determined for each mouse's whole infection period. For this, we determined each individual's area under the curve (AUC) and used these values for comparison. To make this clearer, we rearranged the presentation of the statistics in the figure legend.

3. Fig. 4 – It would be better to calculate geometric mean titers rather than arithmetic – would make difference between IIa and IIb in lung look less different due to outliers in IIa.

We thank the Reviewer for the helpful comment. In Figure 4, we changed the arithmetic mean titres into the geometric mean titres.

4. Fig. 6. Why only 2 data points for IIb in panel B and only 3 in panel C?. Difference in neut antibody appears greater at 12 than 8 days contrary to text that claims early response difference.

This is a valid question, and we fully understand the reviewer's point. In panel b, we only have 2 data points for clade IIb since the amount of serum sample from 2 mice was not enough to additionally run the PRNT assay against clade IIa (panel b) and clade IIb (panel e). We therefore focused on the PRNT against clade IIb, since the animals had been infected with MPXV clade IIb. We only have 3 animals in panel c instead of 7 since this panel presents the antibody responses from 12 dpi. Only 3 out of 7 IIb infected animals reached day 12, while the rest had to be sacrificed due to a humane endpoint. We therefore added the following sentence in the figure legend: "Data from mice that were euthanized prematurely due to humane endpoint assessment are excluded." (lines 728 - 729). We also rewrote the section on the antibody responses on day 8 versus day 12 (lines 210 - 234).

References:

- 1 Tscharke, D. C. & Smith, G. L. A model for vaccinia virus pathogenesis and immunity based on intradermal injection of mouse ear pinnae. *J Gen Virol* **80 (Pt 10)**, 2751-2755 (1999).
- 2 Americo, J. L., Moss, B. & Earl, P. L. Identification of wild-derived inbred mouse strains highly susceptible to monkeypox virus infection for use as small animal models. *J Virol* **84**, 8172-8180 (2010).
- 3 Americo, J. L., Earl, P. L. & Moss, B. Virulence differences of mpox (monkeypox) virus clades I, IIa, and IIb.1 in a small animal model. *Proc Natl Acad Sci U S A* **120**, e2220415120 (2023).
- 4 Gilchuk, I. *et al.* Cross-Neutralizing and Protective Human Antibody Specificities to Poxvirus Infections. *Cell* **167**, 684-694.e689 (2016).
- 5 Yefet, R. *et al.* Monkeypox infection elicits strong antibody and B cell response against A35R and H3L antigens. *iScience* **26**, 105957 (2023).
- 6 Ye, T. *et al.* Polyvalent mpox mRNA vaccines elicit robust immune responses and confer potent protection against vaccinia virus. *Cell Rep* **43**, 114269 (2024).
- 7 Wang, H. *et al.* Rational design of a 'two-in-one' immunogen DAM drives potent immune response against mpox virus. *Nat Immunol* **25**, 307-315 (2024).
- 8 Schroeder, K. & Nitsche, A. Multicolour, multiplex real-time PCR assay for the detection of human-pathogenic poxviruses. *Mol Cell Probes* **24**, 110-113 (2010).
- 9 Ericsson, M. *et al.* Characterization of ts 16, a temperature-sensitive mutant of vaccinia virus. *J Virol* **69**, 7072-7086 (1995).
- 10 Smith, G. L. *et al.* Vaccinia virus immune evasion: mechanisms, virulence and immunogenicity. *J Gen Virol* **94**, 2367-2392 (2013).
- 11 Jones, T. C. *et al.* Genetic variability, including gene duplication and deletion, in early sequences from the 2022 European monkeypox outbreak. Preprint at <https://www.biorxiv.org/content/10.1101/2022.07.23.501239v1> (2022).
- 12 O'Toole, Á. *et al.* APOBEC3 deaminase editing in mpox virus as evidence for sustained human transmission since at least 2016. *Science* **382**, 595-600 (2023).
- 13 Monzón, S. *et al.* Monkeypox virus genomic accordion strategies. *Nat Commun* **15**, 3059 (2024).
- 14 Kramer, T. R. Post-Smallpox Vaccination Skin Eruption in a Marine. *Mil Med* **183**, e649-e653 (2018).
- 15 Ninove, L. *et al.* Cowpox virus transmission from pet rats to humans, France. *Emerg Infect Dis* **15**, 781-784 (2009).

Point-by-point response

We thank the reviewers and the editors for their insightful observations and comments, which have all been answered and enabled us to resubmit an improved manuscript. Below we listed all reviewers' points in black and our answers are in blue. Within the manuscript, edits in response to comments are highlighted in yellow. All editors' points have been answered within the Author Checklist.

Reviewer 1:

This is a revised manuscript. The authors provide an extensive response that adequately respond to reviewers' comments. They make the associated adjustments to the text.

We greatly appreciate the reviewer's positive response.

Minor points to just further enhance the manuscript

1. Line 122/123. During the revisions, authors accidentally switched weight loss after scarification. Fig 3b show clade IIb losing slightly more weight than clade IIa.

We are sorry we missed this mistake and very much appreciate the reviewer's attentiveness. We have now corrected the description about body weight loss for clade IIa and clade IIb. Median body weigh loss was calculated from the individual values across both infection dosages on day 5 (Fig. 3a, b, lines 120 – 122).

2. Line 231. Word choice suggestion for this sentence. Would suggest "below" the detection limit (instead of "beyond" the detection limit)

This is an excellent suggestion, and we have exchanged the word (line 228).

3. Line 770. Typo. VACV (not VACB)

We are sorry to have missed this mistake and very much appreciate the attentiveness of the reviewer. The typo was corrected (line 367).

4. Since Supplementary Table 1 and Supplementary Table 2 just refer to ta scoring system (and not the data collected), I think they should just be referred to once in the main text and then be included in the methods section.

This is a very appropriate suggestion, and we have now removed Supplementary Table 1 and Supplementary Table 2 and included these as scoring in the methods section (lines 95 and 475 – 486).

Reviewer 2:

The authors have responded to my comments in a satisfactorily. The most noteworthy result

is that the authors provide a model for clade II skin infection of CAST mice that is superior to the IN and IP route of infection described previously.

We greatly appreciate the reviewer's positive response.